# Towards routine proteome profiling of FFPE tissue: insights from a 1,220-case pan-cancer study

Johanna Tüshaus [ID][1,9], Stephan Eckert [ID][1,2,3,9], Marius Schliemann[1,4,9], Yuxiang Zhou[2,3,5], Pauline Pfeiffer[5], Christiane Halves[5], Federico Fusco[5], Johannes Weigel[5], Lisa Hönikl[6], Vicki Butenschön[6], Rumyana Todorova[7], Hilka Rauert-Wunderlich[8], Matthew The [ID][1], Andreas Rosenwald[8], Volker Heinemann[7], Julian Holch[7], Katja Steiger[5], Claire Delbridge[5], Bernhard Meyer[6], Wilko Weichert[2,3,5,10], Carolin Mogler [ID][2,3,5], Peer-Hendrik Kuhn[2,3,5] & Bernhard Kuster [ID][1,2,3,4 ✉]

## Abstract

Proteome profiling of formalin-fixed paraffin-embedded (FFPE) specimens has gained traction for the analysis of cancer tissue for the discovery of molecular biomarkers. However, reports so far focused on single cancer entities, comprised relatively few cases and did not assess the long-term performance of experimental workflows. In this study, we analyze 1220 tumors from six cancer entities processed over the course of three years. Key findings include the need for a new normalization method ensuring equal and reproducible sample loading for LC-MS/MS analysis across cohorts, showing that tumors can, on average, be profiled to a depth of >4000 proteins and discovering that current software fails to process such large ion mobility-based online fractionated datasets. We report the first comprehensive pan-cancer proteome expression resource for FFPE material comprising 11,000 proteins which is of immediate utility to the scientific community, and can be explored via a web resource. It enables a range of analyses including quantitative comparisons of proteins between patients and cohorts, the discovery of protein fingerprints representing the tissue of origin or proteins enriched in certain cancer entities.

**Keywords** Clinical Proteomics; Pan-cancer; Mass Spectrometry; TIC Normalization; Public Pan-cancer FFPE Resource
**Subject Categories** Biomarkers; Cancer; Proteomics

## Introduction

Generation of formalin-fixed paraffin-embedded (FFPE) specimens is the standard method prior to long-term storage of patient tissue samples in pathology archives. This cost-efficient preservation method enables storage at room temperature while ensuring tissue integrity for visual or molecular evaluation (Grillo et al, 2017). Apart from standard clinical diagnostics using histology or immunohistochemistry, FFPE sample collections can be used for many purposes in research, particularly when linked to the corresponding clinical data. This may include the discovery or validation of molecular biomarkers by applying e.g. bulk or spatial omics analysis to large patient cohorts. Transcriptomic analysis of FFPE tissue is theoretically viable because high correlation has been observed in paired studies of FFPE and fresh frozen material, but actually remains challenging due to RNA degradation caused by the formalin fixation. RNA isolated from FFPE specimen exhibits lower quality (low median RNA integrity number) than fresh frozen material-derived RNA. This leads to shorter sequencing reads and, in turn, to a higher proportion of unmappable reads (Jacobsen et al, 2023; Skaftason et al, 2022). Because most diseases manifest in altered proteome expression or activity and because most drugs act on proteins, it is conceptually attractive to analyze FFPE material at the proteome level (Coscia et al, 2020; Makhmut et al, 2023; Mund et al, 2022; Welker et al, 2015). FFPE-proteomics has indeed gained traction in recent years owing to a number of important technical advances. First, the challenges associated with protein extraction from chemically cross-linked samples have been substantially addressed by high pressure, high temperature and strong detergent protocols (Buczak et al, 2020). Second, liquid chromatography-tandem mass spectrometry (LC-MS/MS) hardware has become more sensitive, robust and versatile. The latter includes the incorporation of ion mobility-based separation devices such as Trapped Ion Mobility Spectrometry (TIMS) or High Field Asymmetric Waveform Ion Mobility Spectrometry (FAIMS) (Meier

[1]Proteomics and Bioanalytics, School of Life Sciences, Technical University of Munich, Freising, Germany. [2]German Cancer Consortium (DKTK), Partner Site Munich, a Partnership between DKFZ and University Center Technical University of Munich, Munich, Germany. [3]German Cancer Research Center (DKFZ), Heidelberg, Germany. [4]Bavarian Cancer Research Center (BZKF), Munich, Germany. [5]Institute of Pathology, School of Medicine and Health, Technical University of Munich, Munich, Germany. [6]Department of Neurosurgery, School of Medicine and Health, Technical University of Munich, Munich, Germany. [7]Department of Medicine III and Comprehensive Cancer Center Munich, University Hospital, Ludwig-Maximilians University Munich, Munich, Germany. [8]Institute of Pathology, University of Würzburg, Würzburg, Germany. [9]These authors contributed equally: Johanna Tüshaus, Stephan Eckert, Marius Schliemann. [10]Deceased: Wilko Weichert. ✉E-mail: kuster@tum.de

et al, 2018; Swearingen and Moritz, 2012). Third, astonishing improvements in data analysis software have been achieved by incorporating AI-based spectral prediction and chemical modification tolerance into protein identification/quantification search engines (Gessulat et al, 2019; Kong et al, 2017; Yang et al, 2023; Yu et al, 2023). Today, these technical advances enable the measurement of about 5000 proteins from standard-sized FFPE tissue sections (Bhatia et al, 2022; Eckert et al, 2021) or even small tissue areas containing fewer than 100 cells (Makhmut et al, 2023).

FFPE proteome profiling has been applied to the characterization of single cancer types such as colorectal adenomas (Coscia et al, 2020), lung cancer (Friedrich et al, 2021), ovarian tumors (Schweizer et al, 2023) and carcinoma of the esophagus (Li et al, 2023a). Most of these studies comprised relatively small cohorts of up to 100 cases. Larger entity-focused studies are beginning to emerge exemplified by the analysis of matched tumor and benign samples from 278 prostate cancer patients or the analysis of 1780 thyroid nodules (malignant or not) to an average depth of 2500 proteins each (Sun et al, 2022; Zhong et al, 2024). Larger-scale proteome profiling studies including thousands of patients of several entities have so far been limited to fresh frozen tumor samples, most prominently the CPTAC pan-cancer studies (Li et al, 2023b; Savage et al, 2024). Proteomic studies of fresh frozen tissue generally result in deeper proteome coverage and offer the possibility to analyze post-translational protein modifications, however, often suffer from limited specimen availability (Tushaus et al, 2023). To the best of our knowledge, no large-scale pan-cancer FFPE study has been published yet.

We had previously reported a performant workflow (Eckert et al, 2021) that we proposed to be applicable to large-scale proteome analysis of FFPE material but only exemplified it in a few cases. In the current study, we put this workflow to the test in terms of robustness and scalability by analyzing the proteomes of 1220 tumor specimens from six cancer entities (brain, oral, skin, colon, pancreas, lymph node) over the course of three years. Key analytical findings include: (i) demonstrating clear benefits by introducing a new, generic pre-analytical peptide quantification and normalization step to ensure equal and reproducible sample loading for LC-MS/MS analysis across cohorts; (ii) that spiking of retention time standards into patient samples is required to enable close monitoring of chromatographic retention time and the shift thereof across extended periods of time and use of multiple LC columns; (iii) that FFPE tumors can, on average, be profiled to a depth of >4000 proteins within an acceptable timeframe; and (iv) that current software is unable to process such large ion mobility-based online fractionated datasets, requiring the decoupling of several steps of data analysis. The collective data of protein expression of ~11,000 proteins in >1200 tumors from six different entities represents the first comprehensive pan-cancer proteome resource for FFPE material. Initial mining of the data uncovered e.g., tissue of origin-specific as well as cancer entity-specific protein fingerprints. All data is publicly accessible to the scientific community via a custom-built Shiny App enabling a wide range of analyses (https://panffpe-explorer.kusterlab.org/main_ffpepancancercompendium/).

# Results

## Proteomic workflow for profiling a large pan-cancer cohort

Over the course of three years, we processed 1220 patient-derived FFPE tumor tissue samples from six different cancer types, notably glioblastoma (GBM, $N = 246$), oral squamous cell carcinoma (OSCC, $N = 168$), diffuse large B-cell lymphoma (DLBCL, $N = 265$), pancreatic ductal adenocarcinoma (PDAC, $N = 204$), colorectal cancer (CRC, $N = 145$) and melanoma (MEL, $N = 192$) (Fig. 1; Dataset EV1). This pan-cancer cohort was processed and measured in a consistent manner using a workflow previously developed by the authors (Eckert et al, 2021). Briefly, FFPE tumor samples were sectioned and mounted onto cover slides. The tumor area to be analyzed was marked by a trained pathologist, samples de-paraffinized, and the tumor areas of consecutive sections scraped manually into FFPE-lysis buffer. Extensive heating and sonication cycles were applied to aid efficient protein extraction, followed by protein digestion and clean-up using the SP3 approach (Hughes et al, 2019). LC-MS/MS analysis was performed using an Evosep One LC coupled to an Exploris 480 mass spectrometer equipped with a FAIMS unit. Each sample was analyzed twice using an 88 min LC gradient each applying five empirically optimized compensation voltages (CVs) resulting in a total of 10 CVs per tissue sample and a throughput of eight samples per day, totaling 2440 analytical LC-FAIMS-MS/MS runs. Cohorts were analyzed separately in time but with full sample randomization within each cohort. We note that there were considerable gaps in time between different cohorts. In addition, two DLBCL cohorts were included in the study. Analysis of the first ($N = 189$) was separated by 36 months from an add-on cohort (DLBCL+, $N = 76$). This real-world scenario provided a meaningful basis for the assessment of the long-term stability of the workflow. Digests of HeLa cell lysates ($n = 167$ quality control (QC) LC-FAIMS-MS/MS runs) were analyzed at random intervals to monitor system performance.

## Peptide loading normalization by total ion chromatogram calibration

A prerequisite for any quantitative comparison is that equal amounts of peptide/protein are analyzed across the cohort of samples. We found that reliable protein and peptide quantification remains challenging for FFPE tissue extracts due to interferences by e.g. residual paraffin or contaminants such as DNA, lipids or metabolites. The extent of this issue is shown in Fig. 2A where no correlation whatsoever was observed between the results of a total protein assay performed on FFPE protein extracts by a colorimetric assay (660 nm) and a UV-based measurement of the digested peptides (Nanodrop) from the same sample. This likely means that the yield of peptides varies substantially from sample to sample, in turn requiring the addition of a peptide quantification assay to normalize sample quantities across the cohort. To do so, we devised a new method termed TIC normalization which is based on the rational that the total ion chromatogram (TIC) of a peptide sample in an LC-MS analysis should be an accurate representation of the total peptide quantity in said sample and that this quantity can be determined by comparison to a standard of known quantity. Figure 2B illustrates the process in which equal volumes of peptide samples from each patient were analyzed by a short (11.5 min gradient; 100 samples per day (SPD)) LC-FAIMS-MS run (termed Normalization run) to collect the TIC information (see methods for details). Next, a calibration curve was constructed based on a serial dilution of a HeLa cell line digest of known quantity and measured by the same pre-analytical LC-FAIMS-MS method as used for patient samples. It is evident that there is a strict linear relationship between the summed TIC of a sample and the

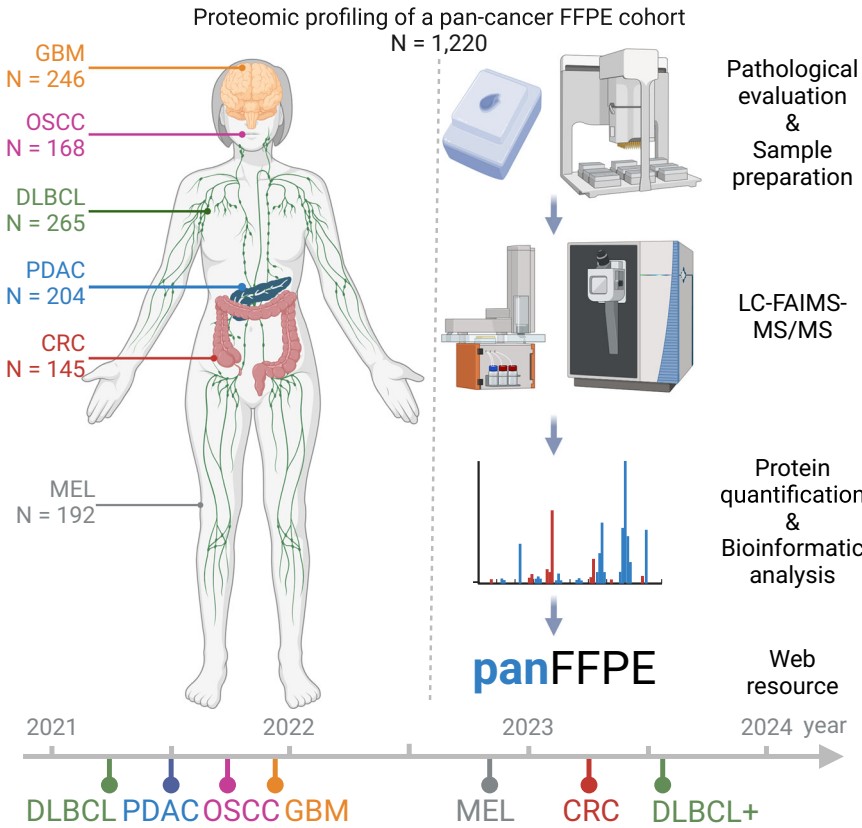

**Figure 1. Study design.**

In total, 1220 FFPE tumor samples from cases of diffuse large B-cell lymphoma (DLBCL), pancreatic ductal adenocarcinoma (PDAC), oral squamous cell carcinoma (OSCC), glioblastoma (GBM), melanoma (MEL) and colorectal cancer (CRC) were proteome expression profiled over a timeframe of three years (starting date of each cohort is indicated) using the workstream shown on the right. The illustration was created with Biorender.com.

amount of peptide analyzed. This allows determining the total peptide quantity in each patient sample and, in turn, adjusting the sample volume for the actual proteomic analysis.

We validated the TIC normalization approach for sample loading normalization by comparing the Qubit protein assay using seven GBM cases. It is apparent from Fig. 2C that more peptides and proteins were identified based on the Qubit assay. However, the TIC normalization led to substantially lower variation between samples at all three levels of assessment (total MS intensity, number of peptides and protein identifications) and was therefore chosen as normalization strategy of our pan-cancer cohort.

Including the TIC normalization step increased the total mass spectrometry instrument time per patient sample by 6.5% but comes with multiple benefits, particularly when analyzing a large number of samples. For instance, applying TIC normalization to all 1424 initial samples in the study, led to the exclusion of 165 samples (11.6%) because of insufficient peptide yield or otherwise poor quality. Hence, we also recommend TIC normalization as a sample triage criterion ahead of the much more time-consuming analytical LC-FAIMS-MS/MS measurement (see Fig. EV1A for example). Removing low-quality samples led to a remarkably low number of cases (6%) where the analytical LC-FAIMS-MS/MS run had to be repeated. Most of these cases were owing to electrospray instability (61%). This is important learning as poor sample quality can rapidly degrade LC column performance, frequent column

changes can lead to more data variation, and sometimes even result in substantial instrument downtime. Applying the TIC normalization approach to all 1,220 remaining samples of the pan-cancer cohort reduced variation in the data by a factor 2–3 within each of the six cohorts and also by a factor >2 across the cohorts (Fig. 2D). Apart from TIC normalization, further quality control measures were implemented. More specifically, we confirmed that peptide yield was independent of the age of the FFPE tumor samples (Fig. EV1B), we spiked synthetic retention time standards (PROCAL peptides) into each FFPE sample (Zolg et al, 2017) (Appendix Fig. S1A) and used HeLa digest samples at random time intervals to track peptide and protein identification rates, quantitative precision and mass measurement error (Appendix Fig. S1B–D).

## Analysis of very large datasets requires decoupling of protein identification, quantification, and FDR control

All attempts to perform peptide and protein identification and quantification along with false discovery rate (FDR) control for all 1220 tissue samples, comprising 2440 analytical LC-FAIMS-MS/MS runs including 5 CVs each in one large analysis failed. Both FragPipe and MaxQuant failed during the label-free quantification step. To overcome this issue, each cohort had to be processed separately. This, however, led to the decoupling of peptide identification/quantification

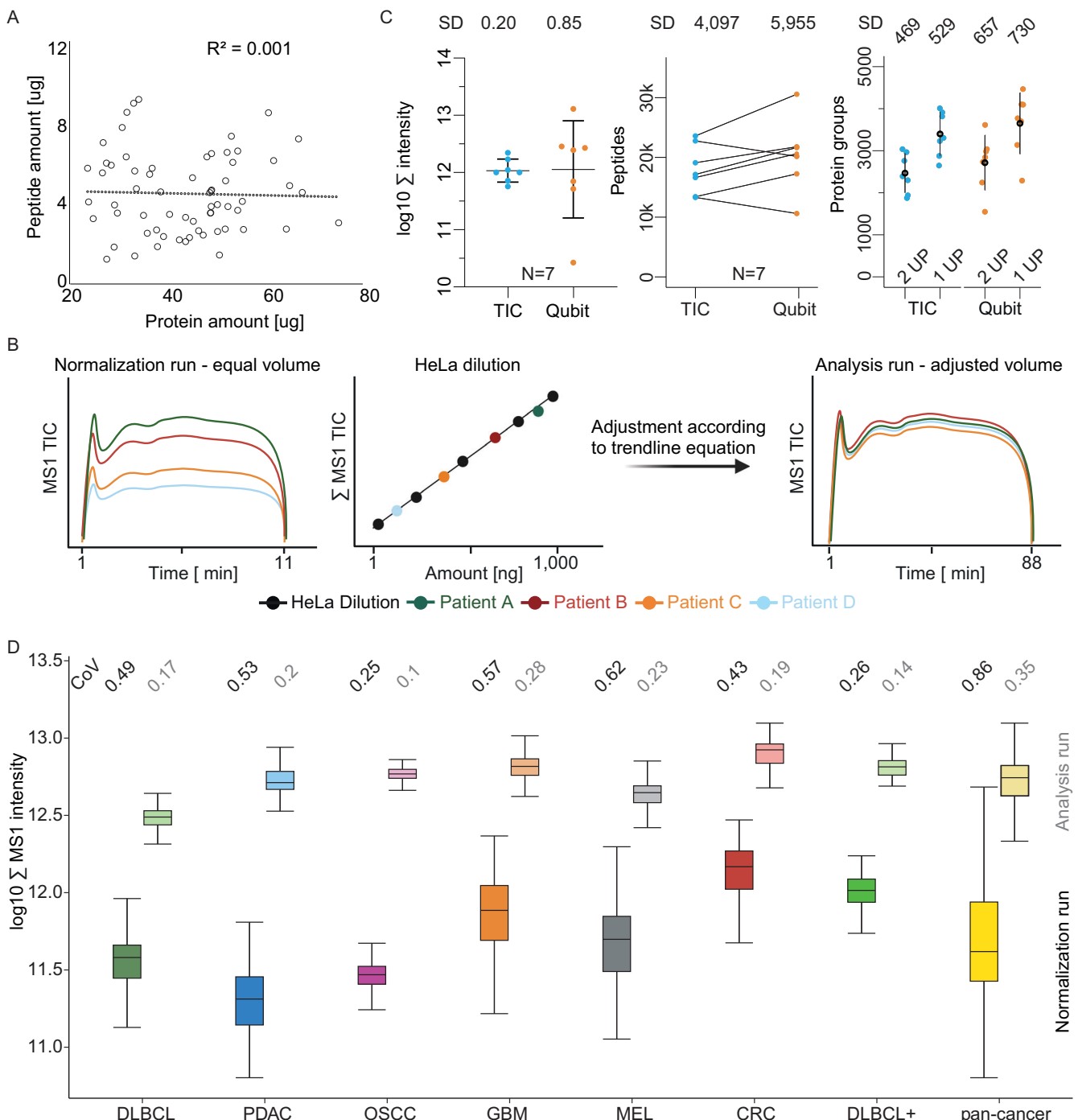

from proper protein grouping and proper FDR control. To deal with this issue, we applied the "picked protein group FDR" software previously developed by the author's group (The et al, 2022) to concatenate the results of the individual cohorts, perform consistent protein grouping as well as FDR control (Fig. 3A).

Next, we explored the effects of different processing options of FragPipe and MaxQuant including post-processing by deep learning on the results regarding proteome coverage and data completeness across the pan-cancer cohort (see methods for details). Five particular combinations were investigated. First,

standard MaxQuant (MQ) (Cox and Mann, 2008), second a combination of MaxQuant with Prosit rescoring (MQ_Prosit) (Gessulat et al, 2019), third standard FragPipe (FP_DDA), fourth FragPipe with MSBooster (FP_Booster) and FragPipe with wide-window acquisition mode (FP_WWA) (Yang et al, 2023; Yu et al, 2021; Yu et al, 2023).

The number of proteins quantified per sample varied substantially within and across the cohorts and also depending on which data processing option was used (Fig. 3B, see Fig. EV2A for peptide level). When requiring at least two unique peptides per protein, MQ

◄ **Figure 2. Sample loading normalization based on total ion chromatograms (TIC).**

(A) Scatter plot comparing the results of protein quantification of FFPE tissue lysate determined by a colorimetric (660 nm) protein assay vs. peptide quantification of respective protein digests using UV absorption (NanoDrop). (B) Schematic illustration of the TIC MS1-only normalization strategy. Equal volumes of each of 1220 peptide digests are analyzed by a 11.5 min LC-MS1 run (left), all signals summed up ($\Sigma$ MS1 TIC) and compared to a calibration curve constructed from a dilution series of HeLa cell digest analyzed in the same fashion (middle), followed by adjusting sample volumes to the same peptide quantity and subsequent analysis by 2 × 88 min LC-FAIMS-MS/MS runs (see methods for details). (C) Comparison of TIC or Qubit sample normalization using FFPE GBM samples ($N = 7$). The determined standard deviation (SD) is consistently smaller for TIC than Qubit for all three metrics applied: summed intensity (left, whiskers and line represent mean ± SD), the number of identified peptides (middle) and the number of identified proteins with one or two peptides, respectively (right, whiskers and dots represent mean ± SD). (D) Boxplots of the summed MS1 intensities of TIC normalization runs (11.5 min) and sample volume adjusted analysis runs (88 min, pale color) for all cohorts. The box represents the interquartile range (IQR) with the lower end being the 1st and the upper the 3rd quartile and its center the median of the underlying data. The whiskers show the span from the box boundaries to the lowest/highest value that is within the range of 1.5 times the IQR, since now outliers are visible this also represents the minimum/maximum value. The numbers on top show the coefficient of variation (CoV) of the TIC sum for each cohort before and after normalization. The sample numbers are 189, 204, 168, 246, 192, 145, 76 and 1220 for DLBCL, PDAC, OSCC, GMB, MEL, CRC, DLBCL+, and pan-cancer, respectively.

quantified a median of 3318 protein groups in each sample and across all samples and cohorts (Fig. EV2B). The numbers for MQ_Prosit, FP_DDA, FP_Booster, and FP_WWA were 3702, 3156, 3250, and 4070, respectively. In each of the cohorts, the average number of quantified proteins was highest for FP_WWA, sometimes by a large margin over others, and was mostly closely followed by MQ_Prosit (Figs. 3B and EV2B). When considering the entire pan-cancer cohort, MQ-Prosit (10,885 proteins) outperformed FP_WWA (10,191) by 7% (Fig. 3C). Reassuringly, 8560 protein groups were quantified by all methods (75% of 11,395 total) regardless of whether the software considered the presence of multiple peptides in one DDA spectrum or not (Fig. 3C). These proteins also had the highest identification scores on average (protein probability of >0.99) compared to proteins only found by four (0.994), three (0.988), two (0.979) or one (0.964) method (Fig. EV2C). Another 760 proteins are supported by four of the five methods (Fig. 3C), suggesting that these are genuine identifications. However, also identifications made just by one method can be of high quality as illustrated by the 686 proteins that were only identified by FP_WWA but with a probability that is nearly indistinguishable from proteins identified by all methods (Fig. EV2C).

The fact that the workflow applied in this study maximizes at about 5000 identified proteins and that the total number of proteins identified across all cancer types is >10,000 already demonstrates the very large differences in protein expression between tissue types (Wang et al, 2019; Wilhelm et al, 2014). This goes hand in hand with the previously published pan-cancer study by the CPTAC consortium using fresh frozen tumor tissue (Appendix Fig. S2). Therefore, one cannot expect data completeness across all cancer types to be very high. Indeed, within a cancer type, the extent of missing data was substantially smaller than across cancer types (Figs. 3D and EV2D). More specifically, even the best-performing FP_WWA method only quantified 25% of all proteins in 77% of all samples (i.e., 23% missingness). All other methods showed substantially poorer performance using this metric. Due to the superiority of FP WWA in terms of completeness and identifications per cohort, all subsequent data analysis was performed using the FP_WWA results.

## Pan-cancer proteome expression profiles distinguish cancer tissues

Collectively, across all samples in a given cancer type, between 9000 and 11,000 proteins were identified and their dynamic expression range spanned 6 orders of magnitude (Fig. EV3A,B). To define a set of proteins that can be used for quantitative comparisons on the

basis of a sufficient number of samples, we systematically assessed the number of quantified proteins as a function of data completeness. The second derivative of the fitted function (change of slope = 0) helped to define a 13% completeness cutoff at which proteins that are quantified in only very few samples would be removed (Fig. EV3C–E, see "Methods" for details). Applying this cutoff to all six cohorts resulted in a total of 7598 proteins that robustly quantified in the pan-cancer cohort with 5081 (67%) proteins being expressed in all entities and 244 being exclusive to one entity only (Fig. 4A). The GBM cohort contained the most exclusive proteins (157), followed by DLBCL (51) and MEL (20) (Fig. 4A).

Next, we used UMAP analysis for a broad comparison of the proteomes of the different cancer cohorts and HeLa QC samples (Fig. 4B). They all formed tight clusters of their constituent patient (or QC) samples and were well separated from the clusters of all other entities. The very tight HeLa cluster is noteworthy as these samples were run along the entire three-year duration of data collection. A very similar observation was made for the proteomes of the first DLBCL and the DLBCL+ cohorts that were analyzed three years apart and the quantitative proteome expression of which were highly correlated (Figs. 4C and EV3F for all other correlations). This is why we merged both DLBCL cohorts for all subsequent analyses. Similarly, we also included healthy lymph node samples (LN, $N = 20$) and healthy oral epithelium samples (OE, $N = 18$) in the analysis and they correlated best with the corresponding cancer entity, namely DLBCL for LN, and OSCC for OE (Fig. EV3F) and were thus also placed in close proximity in the UMAP. Only a few datasets showed outlier behavior, many of which belonged to the MEL cohort, possibly because this cohort contained 30 brain metastasis cases. The above suggests that the separation of cancer entities is primarily driven by protein expression differences between tissues of origin. This is supported by the fact that complete separation was achieved when considering only the top 1000, top 100 and even top 50 (but no longer the top 10) most abundant proteins in an entity (Fig. 4B).

## Differential protein expression between cancer tissues

The current manuscript does explicitly not focus on the biomedical analysis of the data (which will be reported elsewhere). However, the rich dataset provided an opportunity to perform a number of global analyses. Following on from the previous section, we looked into differential protein expression between cancer cohorts. In

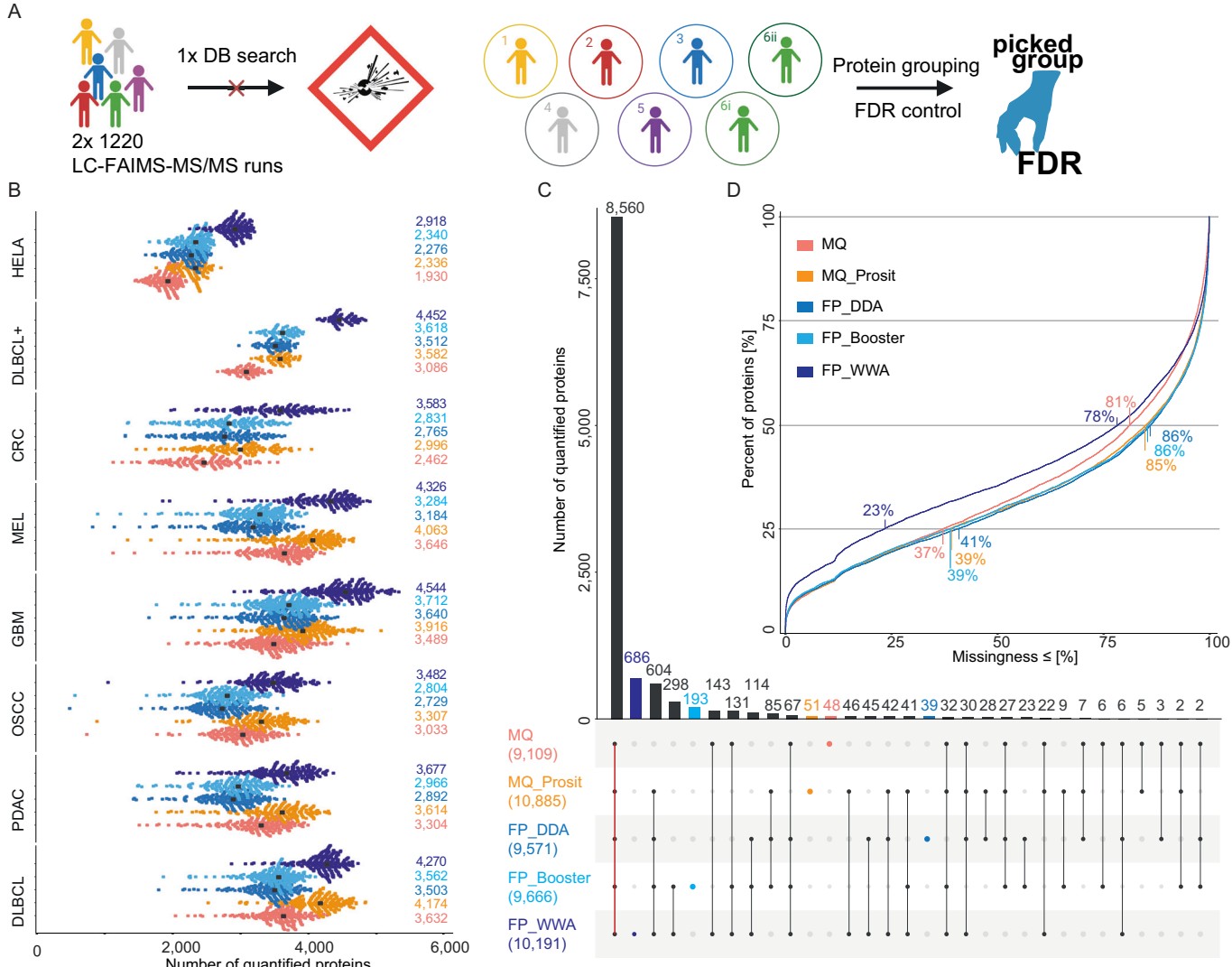

**Figure 3. Comparison of different search strategies and post-processing methods for protein identification.**

(**A**) Illustration of the database search strategy applied to the pan-cancer cohort. (**B**) Swarm plots showing the number of quantified protein groups for each HeLa QC sample and each FFPE tissue sample grouped by cohort and using different search engines and post-processing methods and followed by picked protein group false discovery rate (FDR) control. MaxQuant (red), MaxQuant plus Prosit (orange), FragPipe LFQ workflow without MSBooster (blue), FragPipe LFQ workflow with MSBooster (light blue) and FragPipe wide-window acquisition mode (WWA) (dark blue). The median number of quantified proteins is marked by a black dot and printed on the right. (**C**) Upset plot depicting how quantified proteins are shared between the different searches (colors as in (**A**); the total number of quantified proteins is given in brackets). The bars for and number of proteins exclusively called by one search strategy are highlighted in the respective color. (**D**) Cumulative density plot of the missingness of protein quantifications across all samples from all cohorts, split by the five search strategies and post-processing methods. The percentage of missingness of 25% and 50% of all proteins are indicated.

order to define a meaningful threshold for calling differential protein expression, we randomly assigned samples of each entity into two groups. Calculating the (median) fold change for each protein between these groups showed a maximum absolute log2 fold change of 0.73 by chance alone (Fig. EV4A). We therefore chose this value as the fold change cutoff for all further analysis. To explore differential expression data more systematically, or for particular proteins of interest, we created an R Shiny App (https://panffpe-explorer.kusterlab.org/main_ffpepancancercompendium/) that is publicly accessible. Proteins that are generally more highly expressed in one entity compared to others included well-known cases such as TOP2A, DOCK2, Il16, and BTK for DLBCL and EGFR, CRYAB, and GFAP for GBM (Fig. 5A,B), several of which

are also drug targets in these entities (Appendix Fig. S3). Another example is, TAGLN, reported to be overexpressed in colorectal cancer which indeed showed strong overrepresentation in CRC and PDAC (Fig. EV4B,C) (Liu et al, 2020). In general, we observed a trend for stronger differences for high-abundance proteins and the opposite was true for small differences (Figs. 5A,B and EV4B,C).

A distinct aspect of differential protein expression across cancer cohorts is how often a protein is actually detected within a cohort. For example, the Golgi-resident calcium-binding protein NUCB1 was identified in practically every sample from every cohort and with similar median expression across cohorts, implying house-keeping functions in any cell type (Fig. EV4D). In contrast, while EGFR was detected in 100% of all GBM and 93% of all PDAC cases,

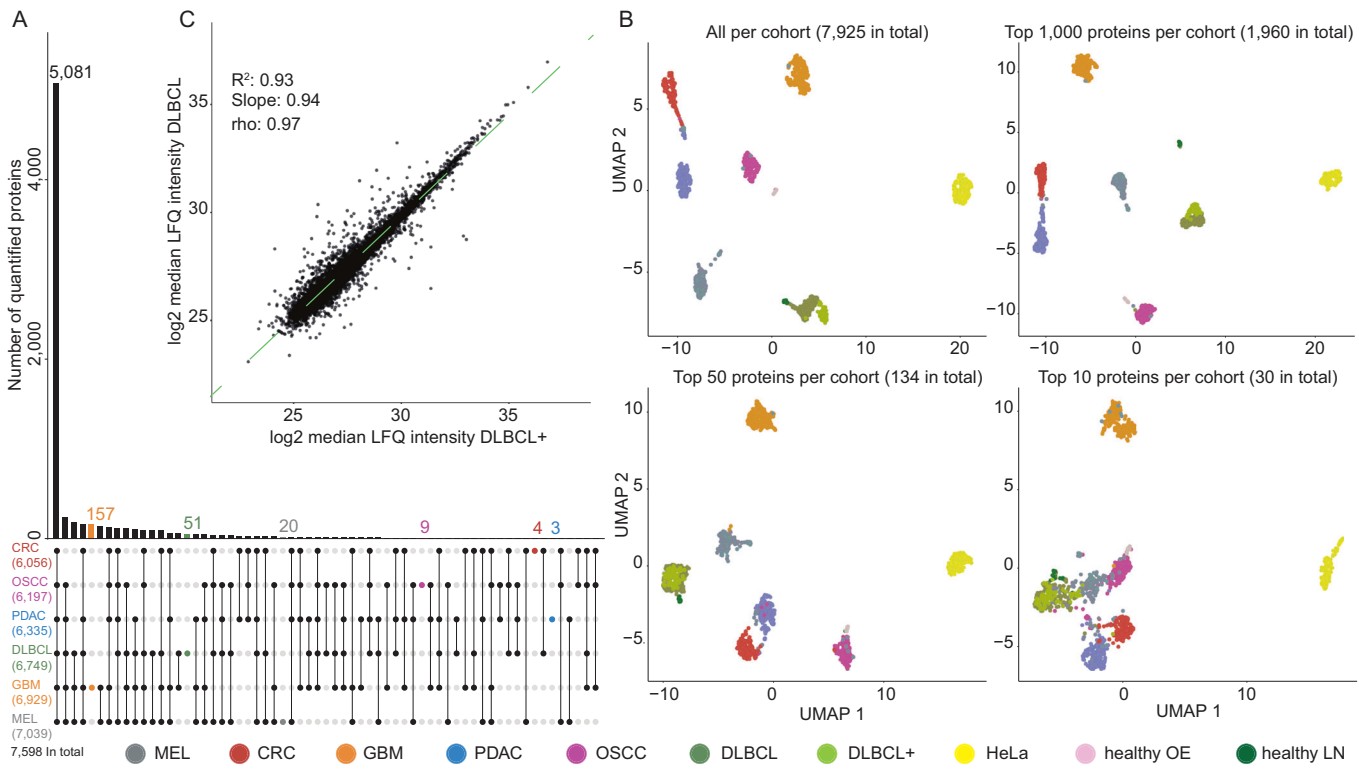

**Figure 4. Global comparison of cancer entity proteome profiles.**

(A) Upset plot depicting how quantified proteins are shared between entities. Proteins were required to be detected in at least 13% of the patients of at least one cohort. Proteins exclusive to one cohort are highlighted in color. The overall number of quantified proteins for each cohort is specified in brackets, as well as the number over all cohorts. (B) UMAP plots of patients clustered on the basis of the abundance of all quantified proteins (top left) or the top N most abundant proteins per cohort. (C) Scatter plot comparing the median log2 label-free quantification (LFQ) intensities of all proteins in the two DLBCL cohorts contained in this study. The green dashed line represents the linear regression fitted to the data ($R^2$: coefficient of determination, rho: Pearson correlation coefficient).

it was only found in 8% of all MEL cases, despite the fact that median expression levels in MEL were very similar to PDAC (Fig. 5B). Another such example is HDAC7 that was detected in 100% of all DLBCL (at high levels) and 74% of all CRC cases but only found in 4% of the OSCC and 9% of the GBM cases even though median expression levels in the latter two entities were similar to that in CRC (Fig. EV4E). These very strong differences are not merely caused by technical variance but imply that some of these rarely detected proteins may, in fact, be related to the cancerous state of that particular tumor. Along the same lines, it is important to note that the expression levels of proteins can also vary within a cohort. Differences between individuals of the same cohort can be very substantial and these can be even larger than median differences across cohorts. Examples include >100-fold expression differences of EGFR in GBM as well as CKB and CDK2 in MEL (Figs. 5B and EV4F). Again, this may imply that extreme expression levels may not be the result of natural variation within a particular cell type but are driven by the cancerous state of a cell.

## Proteomic fingerprints of tissue of origin and of cancer entity

The UMAPs shown in Fig. 4 already demonstrated that the expression levels of a selected number of proteins separated all six

cancer entities. What the maps did not show is if this separation is caused by an underlying proteomic fingerprint that describes the tissue of origin or a proteomic fingerprint describing a tumor entity (or a mixture thereof). To attempt to disentangle the two, we first defined cohort-specific proteomic fingerprints in analogous fashion to the Human Protein Atlas project (HPA, (Uhlén et al, 2015)) but based on the differential expression thresholds determined from our data (see Fig. EV4A). Cohort-specific fingerprints contain three classes of proteins: Class I proteins are exclusively detected in one cohort only, Class II proteins are, on average, 1.66-fold enriched over each individual other entity and Class III proteins show 1.66-fold enhancement in one cohort compared to the average of all other cohorts combined (and not already covered in Class II). Figure 6A shows that the cohort-specific fingerprints of PDAC, OSCC, MEL, and CRC contain substantially fewer proteins than these of GBM and DLBCL.

Next, we investigated to which extent these cohort-specific fingerprints can be explained by tissue of origin or cancer entity. Therefore, we overlaid our data with a previously published study of the HPA project (Uhlén et al, 2015) which defined expression patterns of healthy tissue on RNA level. We chose healthy tissues representing the cancer tissue of origin as best as possible. More specifically pancreas (for PDAC), brain (GBM), salivary gland/ tongue (OSCC), stomach/ intestine (CRC), skin (MEL) and lymphoid tissue (DLBCL). We relied on RNA data because our

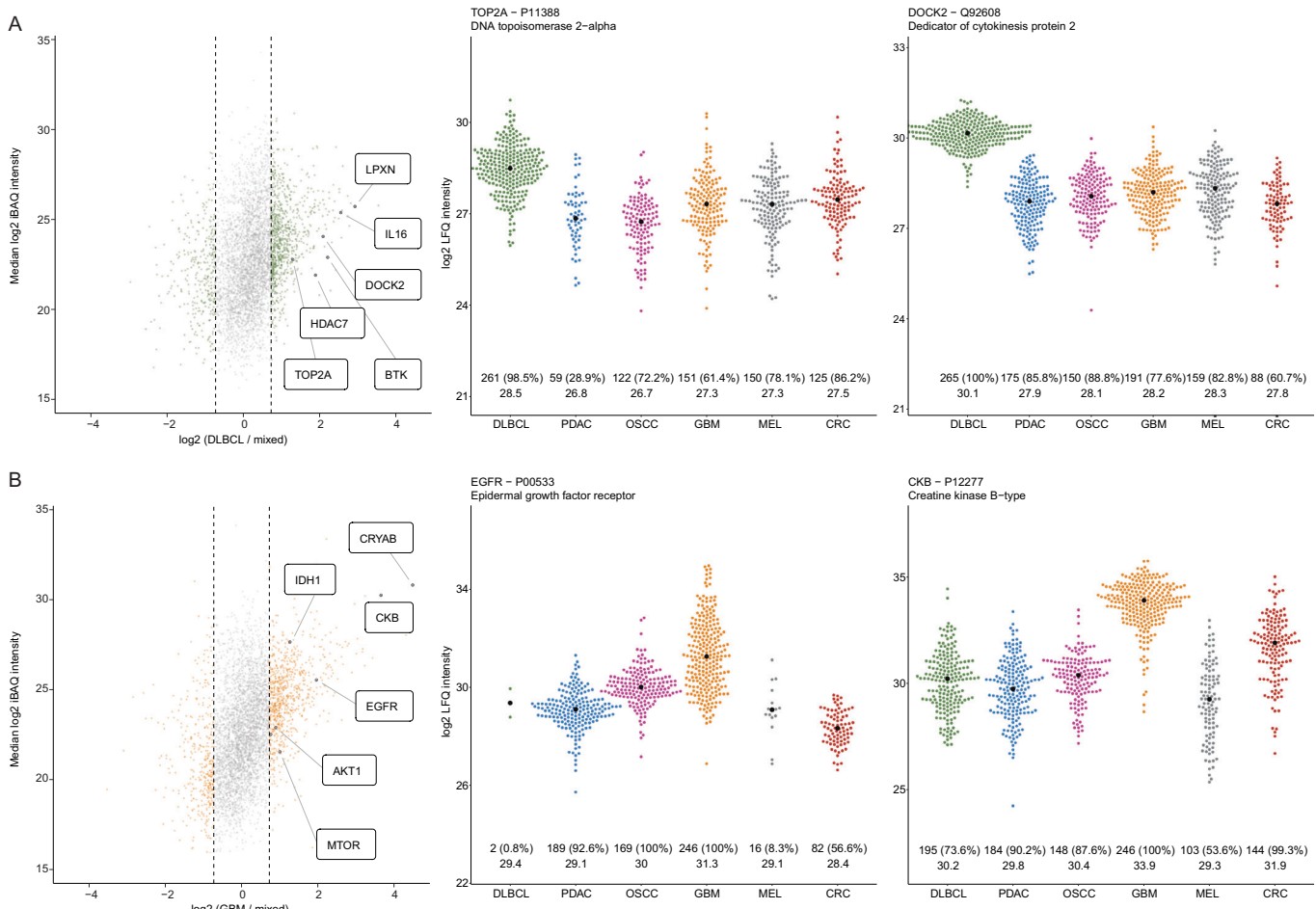

**Figure 5. Quantitative protein expression differences between patients and cancer entities.**

(A) Left: Scatter plot comparing the expression of proteins detected in DLBCL cases (median protein intensity; Y axis) to the (mixed) background of all other entities combined (X axis). The dashed lines represent the chosen fold change cutoff of ±0.73 (see "Methods"). Middle and right: swarm plots showing the expression of exemplary proteins (each dot is a patient) enriched in the DLBCL cohort compared to all other cohorts. Numbers at the bottom indicate the number of patients the protein was detected in (N), the percentage of patients in which the protein was detected within the cohort in brackets and the median LFQ intensity of this protein in each cohort. (B) Same as (A) but for GBM.

previously published proteomic data on healthy human tissues (Wang et al, 2019) did not contain all the tissues covered in the current study. According to this analysis, only a minority of the proteins that make up the cohort-specific fingerprint appear to reflect the tissue of origin. Very few such proteins were found for PDAC, MEL and CRC (2–3% of the cohort-specific fingerprint), followed by OSCC (18%) DLBCL (19%) and GBM (21%) (Fig. 6B). In absolute terms, DLBCL and GBM stood out with >200 proteins comprising the tissue-specific fingerprints. This is consistent with brain and lymphoid tissues being, together with testis, the three tissues known to show the most distinct expression patterns in the human body (Uhlén et al, 2015).

We detected 112 proteins annotated as oncogenes as well as 93 proteins annotated as tumor suppressor genes (according to UniProt) in the entire pan-cancer cohort, some showing quantitative expression differences between cohorts (Appendix Fig. S3). Of the latter, 39 were part of a cohort-specific fingerprint including RPS6KA2, CADM4, and NF1 for GBM as well as ATM, BAX, RB1,

and RASSF5 for DLBCL (Fig. 6B). Similarly, 52 oncogene-encoded proteins were also fingerprint proteins including MYH11 for CRC, FAM83B for OSCC, HMGA2 for MEL, and BCL2, BCL6, ELL, RAB8A, and DDX6 for DLBCL. Importantly, some of the oncogene-encoded proteins enriched in DLBCL (LCK, LYN, HCK, VAV, OBF1) and GBM (KPCA, OLIG2) are also enriched in their tissue of origin (lymph node and brain, respectively) (Fig. 6B). Therefore, these expression levels of these proteins are unlikely to represent a cancerous state of a cell.

To learn more about what proteins and molecular functions constitute and distinguish cohort-specific, tissue of origin-specific, and cancer entity-specific fingerprints, we performed gene ontology (GO) term enrichment analysis for all three categories (Fig. 6C; Appendix Fig. S4A,B). As one might expect, standard GO analysis for molecular function, biological process or cellular localization highlighted many terms characteristic for the tissue of origin. For instance, secreted proteins were particularly high for PDAC and OSCC, both tissues/glands whose intrinsic function is to secrete a

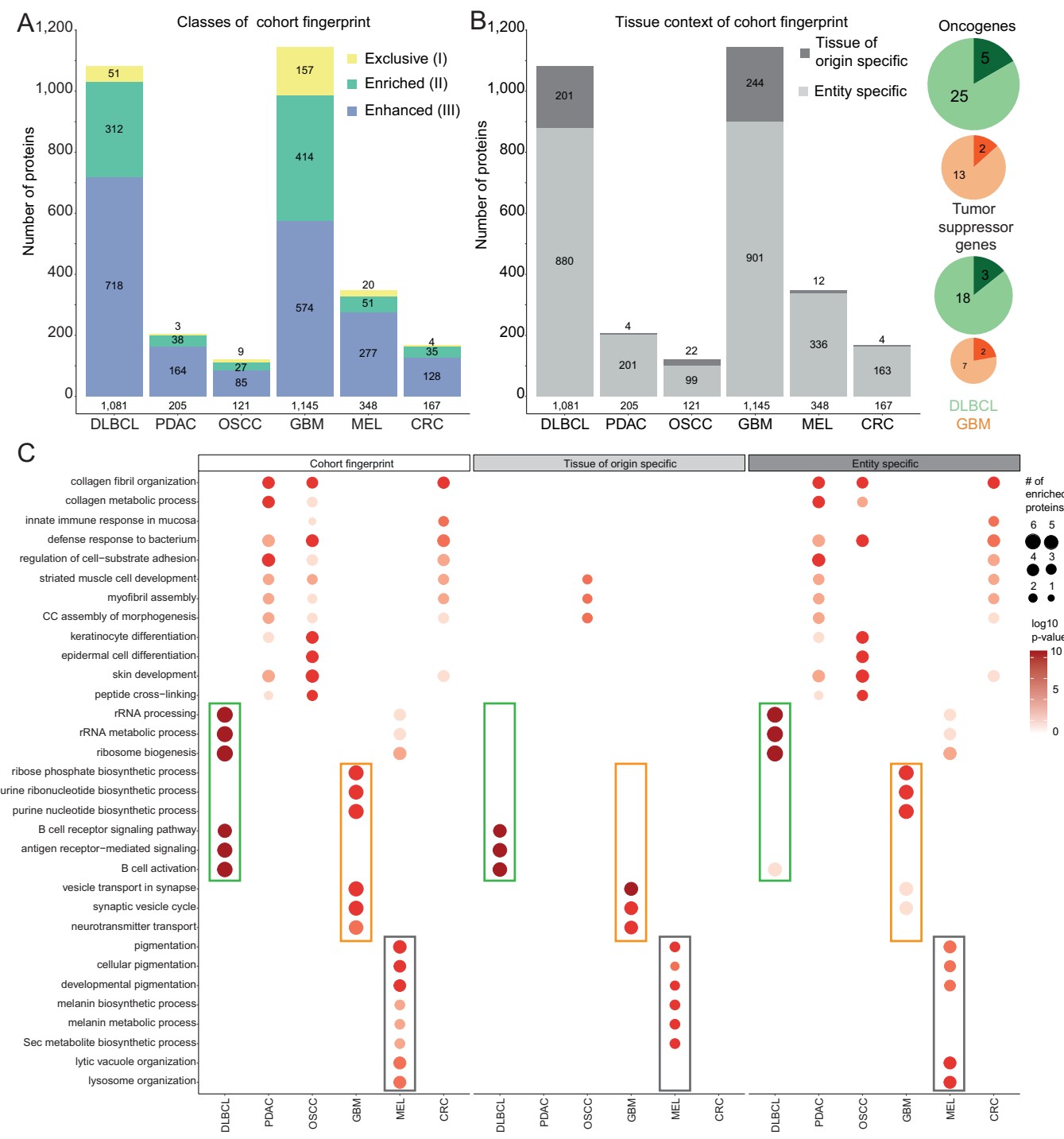

**Figure 6. Proteome fingerprints of tissue of origin and cancer entity.**

(**A**) Bar graph of the number of proteins forming cohort-specific fingerprints: Class I—exclusive proteins: uniquely expressed in one cohort only. Class II—enriched proteins: at least 0.73 log2 fold change over the median of each individual other cohorts. Class III—enhanced proteins: at least 0.73 log2 fold change over the median of all other cohorts combined (see methods). (**B**) Same as A) but split by tissue of origin specificity (based on RNA-seq profiles from Uhlén et al, 2015) and cancer entity specificity (this study). Pie charts indicate oncogenes and tumor suppressor proteins (TSP) detected as cohort-specific fingerprint proteins in DLBCL or GBM. Darker color again indicates the fraction of tissue of origin-specific proteins of the fingerprint. (**C**) Gene ontology (GO) term enrichment analysis (biological process) of the cohort fingerprints, tissues of origin and cancer entity-specific proteins for all cohorts. The dot size represents the number of enriched proteins for the given GO term and the color scale indicates the statistical significance of the enrichment. Frames highlight examples that are tissue of origin or entity-specific or attributable to both.

lot of proteins in the gut and oral cavity, respectively. However, and interestingly, while B-cell receptor signaling was enriched in the DLBCL cohort- and tissue of origin-specific fingerprints, it was absent from the entity-specific fingerprint. Likewise, melanin biosynthesis scored in the MEL cohort but not in the entity-specific fingerprint for MEL. Similarly, GO terms related to B-cell activation were statistically significant in the tissue of origin but not the cancer entity-specific fingerprint of DLBCL. Instead, transcription-related processes were overrepresented in the entity-specific fingerprint of DLBCL and, to a lesser extent in MEL, although proteins from the nucleus, the cellular compartment where these processes happen, are not systematically higher abundant for these cohorts (Appendix Fig. S4C). We also employed an enrichment analysis on a database published by the CPTAC consortium that focuses on cancer hallmarks (Liberzon et al, 2015) and these were indeed more densely represented in the entity-specific fingerprints than the tissue or origin fingerprints (Fig. EV5). The above analysis suggests that it is principally possible to distinguish cancer-related proteome expression from that of the underlying tissue of origin.

## Discussion

In this study, we compiled the first comprehensive repository of pan-cancer proteomics data obtained from clinical archived FFPE material. The resource collectively contains protein expression information for nearly 11,000 proteins as a result of analyzing the proteomes of >1200 cancer cases (Fig. 1). The primary purpose of assembling the data in this way was to assess if our previously published workflow (Eckert et al, 2021) would scale to larger numbers of samples and would be robust over extended periods of time. In terms of scale, the current implementation of the workflow realistically enables the analysis of about 2500 patients per LC-MS/MS system per year. In terms of robustness, the achieved median quantitative precision for one cancer cohort was between 10–28% coefficient of variation (CoV) and only slightly worse (35% CoV) across all cohorts and a time span of more than three years. These two key metrics suggest that the workflow would be able to support even phase 3 or phase 4 clinical trials with high-quality quantitative data for thousands of proteins in parallel.

We found that the most important pre-analytical factor for achieving this level of quality was to ensure equal peptide sample loading onto the mass spectrometer (Fig. 2). This requirement was delivered by introducing a novel but simple mass spectrometry-based normalization approach that uses the TIC of rapid LC-MS analyses of each patient sample compared to an external standard (HeLa cell digest). This TIC normalization step was also an effective means to remove samples of poor quality from subsequent steps of the process, thus saving time and improving overall data quality. A conceptually similar approach has been published before that makes use of the extracted MS2 signal intensity of identified peptides in a sample (Makhmut et al, 2023). The advantage of our TIC normalization method is that it is more generally applicable because it does not require any peptide or protein identification information. This can be advantageous when analyzing morphologically very different tissues or differences in protein extraction, storage time or tumor cell content.

An important analytical factor was to include synthetic peptide standards into each sample that were used to monitor the sometimes substantial drifts in chromatographic retention time behavior of patient proteomes over time. These shifts may result from e. g. column deterioration, column changes or sample matrix effects and can complicate retention time recalibration performed by the search engines. The idea is not new (Holman et al, 2016) but a distinct advantage of our retention time standard (PROCAL; Zolg et al, 2017) over others is that it contains many peptides ($n = 40$) spread across the whole gradient, enabling a seamless monitoring over the whole retention time range and increasing the chances of robustly detecting a sufficient number of peptides in every sample. Mass stability was found to be a minor issue because modern software packages re-calibrate peptide masses on the basis of identified peptides as one of the steps of data processing resulting in a median mass accuracy of <1 ppm without the need for added standards.

All data was acquired in the so-called data-dependent acquisition (DDA) mode of operating a mass spectrometer (see also below). We identified a somewhat surprising post-analytical bottleneck such that both software tools used in this study (MaxQuant and FragPipe) were unable to process all samples in one large analysis even though a high-performance server computer was used (512 GB of RAM and 128 cores; Fig. 3). Both failed at the label-free quantification step. This is likely due to the large number of data files that had to be processed. The patient samples alone created 2440 raw data files and MaxQuant required splitting these up into five parts corresponding to one FAIMS compensation voltage each (totaling 12,200 raw data files). This issue was overcome by performing protein identification and quantification separately for each cohort which reduced the number of raw MS files to be processed to ~150–500 and followed by a consolidation and FDR-controlled protein grouping step across all cohorts using the picked protein group FDR tool (The et al, 2022). While providing a solution to the problem at hand, this decoupling is at least inconvenient (if not disabling) for scientists who lack computational support and, therefore, rely on using software tools as provided by academic or commercial developers. The authors note that the extent to which the data has to be divided into manageable parts will also depend on the computational power allocated for processing. As the field is aspiring to analyze very large numbers of samples, just adding compute power to whole project-level data processing is not going to scale. Therefore, we think that future data processing tools will have to implement a pipeline or streaming process in which the heavy data processing steps are performed as the data is produced so that the final consolidation step across all samples is still computationally tractable. On a more positive note, the application of AI-based tandem mass spectrum prediction tools led to more proteins and more data consistency for both search engines used in this study—MaxQuant-Prosit and MSFragger-MSBooster (Gessulat et al, 2019; Yang et al, 2023; Yu et al, 2021; Yu et al, 2023), the latter delivering the best overall performance.

Further future directions in the overall workflow development may include improvements in sample throughput and sensitivity enabled by the latest generation of mass spectrometers (such as the Orbitrap Astral and timsTOF ultra (Ctortecka et al, 2024; Heil et al, 2023) that may push the number of patient samples to >10,000 case

per year per instrument and improve depth to >6000 proteins per case. This would help to alleviate a substantial current challenge in clinical proteomics i. e. making sure that data can be collected at comparable depth and quality in different laboratories. Such multi-center studies are desirable in principle because the number and types of cancer cases that can be subjected to proteomic profiling may be increased. However, multi-center studies often suffer from the introduction of lab-specific batch effects and designing and enforcing standard operating procedures across sites remains difficult. An alternative approach would be to establish centralized laboratories at clinical centers with a particularly strong focus and expertise on certain cancer entities for an entire country or region. This would also require a way to coordinate the sample flow from decentralized biobanks to centralized expert-guided proteome centers.

Another area of future development is data consistency enabled by data independent acquisition (DIA). We consciously decided against a DIA approach at the beginning of the project for two main reasons. First, we expected that the overall proteome profiles of the pan-cancer cohorts would be quite different. Hence, it was unclear how one would design high-quality spectral libraries that would adequately capture this anticipated (and now experimentally confirmed) heterogeneity and, at the same time, control for false positive assignments of proteins that, in fact, are only expressed in a subset of all samples and thus not missing for technical reasons. Again, the very fast and sensitive new-generation mass spectrometers now enable narrow-window DIA measurements that diminish this particular issue. In addition, the scalability of DIA data analysis software was also unclear at the time and substantial improvements in DIA software performance are still needed in order to cope with the anticipated higher and higher data loads. Going forward, the authors think that proteome analysis software will have to move away from batch-wise processing of samples at the end of a project, but instead develop into a streaming mode of operation in which newly acquired data can be analyzed immediately, yet still aggregated as time progresses. As a result of project decisions taken several years ago, the current dataset is not perfect in terms of proteomic depth (i.e., comprehensiveness) and data completeness across samples and cohorts, but it should be well controlled for false discoveries.

The assembled dataset presented an opportunity to explore proteome expression differences across cancer entities (Figs. 4 and 5). As observed before for healthy human tissue (Wang et al, 2019; Wilhelm et al, 2014), the quantitative expression of few (50–100) high-abundance proteins was sufficient to distinguish the different cancer entities in a UMAP analysis, suggesting a strong influence of the underlying tissue of origin. This may seem obvious, but it is still very useful information when analyzing differences in protein expression within a cancer cohort or across cancer entities because certain (e. g. proto-oncogenes such as EGFR) may be expressed at very different levels in different cell types (cancerous or not). At the same time, it also became clear that the oncogenic transformation of tumor cells leads to widespread changes in overall proteome expression which cannot just be explained by the tissue of origin (Fig. 6). Because only a few ways of making use of the assembled data could be covered in this report, we have built an interactive tool (https://panffpe-explorer.kusterlab.org/main_ffpepancancercompendium/) that we anticipate the scientific community will find useful for further exploring and benefiting from the data.

# Methods

## Reagents and tools table

| Reagent/resource | Reference or source | Identifier or catalog number |
|---|---|---|
| **Experimental models** | | |
| Human HeLa cell line CCL-2 | ATCC | – |
| **Chemicals, enzymes, and other reagents** | | |
| Trypsin | Roche | Lab stock |
| SP3 beads | Merck | SPEEDBEAD MAG CARBOXYL MODIF GE45152105050350 GE65152105050350 |
| 2-Chloroacetamide (CCA) | Sigma-Aldrich | C0267 |
| PROCAL -Retention Time Standardization Kit | JPT | Zolg et al, 2017 |
| Pierce 660 nm Protein Assay Reagent | Thermo Fisher Scientific | 22660 |
| tris(hydroxymethyl) aminomethane (TRIS) | Merck | 1.08382 |
| sodium dodecyl sulfate (SDS) 20% in water | Sigma | 05030 |
| Qubit Protein Assay-Kit | Thermo Fisher Scientific | Q33211 |
| **Software** | | |
| MaxQuant | Cox and Mann, 2008 | – |
| FragPipe | Yu et al, 2020; Chang et al, 2020; Kong et al, 2017 | FragPipe v21.1, MSFragger v4.0, IonQuant v1.10.12, Philosopher v5.1.0, DIANN v1.8.2_beta_8 |
| Prosit | Gessulat et al, 2019 | – |
| Picked group FDR tool | The et al, 2022 | – |
| R | Posit PBC | v4.1.0 |
| R Studio | The R Project | v2023.12.0 + 369 |
| Xcalibur | Thermo Fisher Scientific | v4.4 |
| Tunes | Thermo Fisher Scientific | v2.0 |
| **Instrumentation** | | |
| Evosep One | Evosep | EV1000 |
| EvoTips | Evosep | EV2001 |
| Stainless steel emitters | Evosep | EV1086 |
| Endurance column 8 cm length, 100 µm inner diameter, 3 µm particle size | Evosep | EV1064 |

| Reagent/resource | Reference or source | Identifier or catalog number |
|---|---|---|
| Endurance column 15 cm length, 150 μm inner diameter, 1.9 μm particle size | Evosep | EV1106 |
| FAIMS Pro | Thermo Fisher Scientific | FMS02-10001 |
| Orbitrap Exploris™ 480 mass spectrometer | Thermo Fisher Scientific | BRE725533 |
| Agilent Bravo Automated Liquid Handling Platform | Agilent | G5563AA |

## Tissue samples

The FFPE tissue samples used in this study were acquired at different pathological institutes. Melanoma (MEL), glioblastoma (GBM), oral squamous cell carcinoma (OSCC), pancreatic ductal adenocarcinoma (PDAC) and the first batch of diffuse large b-cell lymphoma (DLBCL) FFPE specimens were collected at the Institute of Pathology at the University Hospital rechts der Isar (MRI TUM). A second batch of DLBCL FFPE cases (DLBCL+) were collected at the Institute of Pathology in Wuerzburg, whereas the colorectal cancer (CRC) cohort was processed by the LMU in Munich. The investigation was approved by the local ethical committee of the responsible University Hospitals (TUM: 33/20 S, 296/17 S, 532/16 S, 2024-60-6-KK, LMU: 2006-004030-32, Wuerzburg: 149/23). The OSCC and the DLBCL cohort included healthy reference samples ($n = 18$ healthy oral epithelium and $n = 20$ healthy lymph nodes, respectively). For each FFPE block, a hematoxylin and eosin staining was prepared. This reference slide was used for the annotation of tumorous tissue by a trained pathologist.

## Protein extraction from FFPE tissue

In all, 10-μm FFPE tissue sections were cut, using a microtome, and mounted onto glass slides for subsequent deparaffinization. First, slides were incubated at 60°C for 30 min and then dipped into 100% xylene for 20 min to remove bulk paraffin. Next, the tissue was rehydrated for 10 min in each solvent in an ethanol (EtOH) gradient, following the order 100% EtOH, 96% EtOH, 70% EtOH and finally 3-times deionized water.

From each slide, tumorous tissue was dissected using a scalpel and transferred into lysis buffer. Lysis buffer consisted of 100 mM Tris-HCl (pH 8) containing 4% sodium dodecyl sulfate (SDS) and 10 mM dithiothreitol (DTT) for DLBCL, OSCC, PDAC, and DLBCL+ and 500 mM Tris-HCl (pH 9) containing 4% SDS and 10 mM DTT for GBM, CRC, and MEL. Proteins were extracted by boiling for 1 h at 100°C. Next, extracts were sonicated using the Bioruptor Pico (Diagenode, for 25 cycles at 8°C, 60 s on and 30 s off) or the FFPE protocol of the Covaris Sonicator, debris was pelleted by centrifugation at 17,000 × $g$ for 10 min, and the supernatant was further processed by SP3 digestion (Hughes et al, 2019).

## Preparation of HeLa peptides for quality control

Human HeLa cells (ATCC CCL-2) were cultured in Dulbecco's Modified Eagle Medium (DMEM) containing 10% Fetal Bovine Serum at 37°C and 5% $CO_2$. Cells were lysed by incubation in lysis buffer (8 M urea, 1× Ethylenediaminetetraacetic acid (EDTA) free protease inhibitor in 40 mM Tris-HCl, pH 7.6) on ice for 5 min. The lysate was cleared by centrifugation for 30 min at 4°C at 20,000 $g$. Proteins were reduced with 10 mM DTT for 45 min at 37°C and alkylated with 55 mM 2-chloroacetamide (CAA) for 30 min at room temperature. Samples were diluted to 1 M urea using 40 mM Tris-HCl, pH 7.6. Initial digestion of proteins was performed by adding Trypsin and Lys-C (1:100 (wt/wt) enzyme-to-protein ratio) at 37°C for 4 h. A second addition of the same amount of both proteases was performed and the digestion was allowed to continue overnight. Protease digestion was stopped by the addition of formic acid (FA) to a final concentration of 1%. The acidified peptides were centrifuged at 5000 × $g$ for 15 min, followed by desalting on a Sep-Pak C18 Cartridge. The peptide concentration was estimated by NanoDrop measurements.

## Protein clean-up and digestion of FFPE material

The protein concentration in the protein extract from FFPE tissue was determined using the Thermo Pierce 660 nm protein assay. For compatibility with SDS, 50 mM alpha-cyclodextrin was added. All steps were performed according to the manufacturer's protocol.

Single-Pot Solid-Phase-Enhanced sample preparation (SP3) was conducted as described by Hughes et al (Hughes et al, 2019) to remove SDS and other contaminants prior to enzymatic digestion. GBM, CRC and DLBCL+ cases were processed in the 96-well plate format on a Bravo Agilent robotic liquid handling platform, whereas for the other cohorts the SP3 procedure was performed manually.

For automated processing, 50 μg protein extract was first mixed with SP3 beads (50:50 mixture of Sera-Mag carboxylate-modified magnetic bead types A and B (Merck)) in a deep well plate to a 1:5 protein-to-bead ratio. Proteins were precipitated onto the bead surface at 70% EtOH, washed twice with 80% EtOH, and once with 100% acetonitrile (ACN). After the clean-up, beads were resuspended in 50 μl digestion buffer, containing 10 mM tris (2-carboxyethyl)phosphine (TCEP) and 2 mM $CaCl_2$ in 40 mM Tris-HCl at pH 7.8 and incubated for 45 min at 37°C. Reduced cysteines were alkylated for 30 min at room temperature using 55 mM CAA. Trypsin was added in a 1:50-ratio (enzyme-to-protein, (wt/wt)), and enzymatic digestion continued overnight at 37°C. The next day, beads were settled using a magnet and the supernatant was either manually or automatically transferred to a tube or plate, respectively. Beads were washed with 50 μl 2% FA, supernatants combined, and acidified if the pH was above 3.

If processed manually, the protocol was performed in tubes, 60 μg of protein was used and precipitation was induced by 70% acetonitrile. In addition, Lys-C and Trypsin were used in a two-step digestion, each added in a ratio of 1:100 (wt/wt), and the same amount was added again after 4 h.

## Evotip loading

To desalt and prepare peptides for LC-MS(/MS) analysis, the digest was loaded onto Evotip disposable trap columns (Evosep)

**Table 1. Detailed information on MS methods.**

| Sample type | | TIC normalization | HeLa QC | Clinical FFPE specimens |
|---|---|---|---|---|
| Peptide amount | | to be determined; constant volume | 300 ng | 2×600 ng ( + 200 fmol PROCAL) |
| Gradient length | | 11.5 min (100 SPD) | 44 min (30 SPD) | 2×88 min (2×15 SPD = ~8 SPD) |
| Flow rate | | 1500 nl/min | 500 nl/min | 220 nl/min |
| Column | | 8 cm, 3 µm particle size, 100 µm ID | 15 cm, 1.9 µm particle size, 150 µm ID | 15 cm, 1.9 µm particle size, 150 µm ID |
| FAIMS CVs | | −45 V\|−65 V | −45 V | set1: −30\|-40\|−50\|−60\|-70 V<br>set2: −35\|− 45\|-55\|−65\|−75 V |
| Cycle time | | – | 2 s | 0.6 s |
| MS1 | Scan range | 360–1300 *m/z* | 360–1300 *m/z* | 360–1300 *m/z* |
| | Resolution | 120k | 60k | 60k |
| | maxIT | 45 ms | 45 ms | 45 ms |
| | AGC | 100% (1E6) | 100% (1E6) | 100% (1E6) |
| MS2 | Isolation window | – | 1.3 Th | 1.3 Th |
| | Charge state | – | 2–6 | 2–6 |
| | Dynamic Exclusion | – | 45 s | 90 s |
| | Resolution | – | 15k | 15k |
| | maxIT | – | 25 ms | 25 ms |
| | AGC | – | 100% (1e5) | 100% (1e5) |

according to the manufacturer's protocol with slight changes. All washing and loading volumes were increased from 20 to 100 µl such that Evotips were equilibrated in three steps: (i) 0.1% FA in acetonitrile (EvoB); (ii) 30% EvoB in EvoA; and (iii) 0.1% FA in water (EvoA). Prepared tips are best stored in EvoA at 4°C, preventing them from running dry.

## Estimation of peptide concentration

### Qubit
Estimation of peptide concentration directly in the peptide digest using the Qubit assay (Thermo Fisher Scientific) was performed as indicated in the manufacturer's protocol.

### Nanodrop
To estimate the peptide concentration after SP3 clean-up and digest, samples were desalted using C18 Stagetips as described by (Rappsilber et al, 2007) prior to spectroscopic absorbance measurement.

### TIC normalization
For the TIC normalization approach, a small constant fraction of the total digest volume was loaded onto Evotips (see above) for each sample, along with an 8-point HeLa dilution series ranging from 16 ng to 1000 ng peptide per tip. The samples were analyzed via LC-FAIMS-MS using 2 CVs (−45 and −65 V) and an 11.5 min (100 samples per day (SPD)) gradient (see Table 1 for details). As an approximation of the loaded peptide amount the total ion current (TIC) of all MS1 scans was summed up (equivalent to the area under the TIC chromatogram). Using this metric of the HeLa dilution series with known peptide amount a linear calibration curve was generated. R2 of the HeLa dilution curve needed to be above 0.97 to ensure high quality. If this quality criterion were not fulfilled, measurements were repeated with a freshly prepared HeLa dilution series. Based on this regression the injected amount and in

turn the concentration in the peptide digest of the FFPE samples was calculated. The R script used for this purpose along with exemplary output files is available at the kusterlab GitHub page (https://github.com/kusterlab/TIC_Normalization).

### NanoLC-FAIMS-MS/MS analysis
NanoLC(nLC)-FAIMS-MS and nLC-FAIMS-MS/MS measurements were performed on an Evosep One LC-system (Evosep) coupled to an Orbitrap Exploris 480 mass spectrometer (Thermo Fisher Scientific). The mass spectrometer was equipped with a FAIMS Pro unit (Thermo Fisher Scientific) to utilize ion mobility-based fractionation. Tip-bound peptides were eluted using different by the vendor predefined low-pressure gradients of 11.5 min (100 SPD), 44 min (30 SPD) and 88 min (15 SPD) at flow rates of 1500, 500, and 220 nl/min, respectively. For TIC normalization a fixed volume of peptide digest was analyzed (see Estimation of peptide concentration—TIC normalization) using the 100 SPD method, peptides were refocused and separated on a Evosep C18 column (8 cm, 3 µm particle size, 100 µm inner diameter (ID)). As interspersed quality controls 300 ng of HeLa peptides and for the in-depth measurement of clinical FFPE tissue 2 × 600 ng of peptides plus 200 fmol synthetic peptide retention time standard each (PROCAL, (Zolg et al, 2017)) were separated on a 15 cm C18 column (1.9 µm particle size, 150 µm ID) at a throughput of 30 SPD and 15 SPD, respectively. The mass spectrometer was operated in positive ionization mode with a spray voltage of 2300 V. The FAIMS unit was set to standard resolution (inner and outer electrode 100°C). For the 100 SPD method 2 CVs (−45 and −65 V), for the 30 SPD method 1 CV (−45 V), and for the 15 SPD method 2×5 CVs (set1: −30\|− 40\|−50\|-60\|−70 V, set2: -35\| −45\|−55\|−65\|−75 V) were chosen (Eckert et al, 2021).

The full scans (MS1) were acquired over a mass-to-charge (*m/z*) range of 360–1300 and the resolution was set to 120,000 at *m/z* 200 (100 SPD) and 60,000 at *m/z* 200 (30 SPD and 15 SPD) with a

maximum injection time (maxIT) of 45 ms and a normalized AGC target value of 100% (1E6).

For the 100 SPD gradient, only MS1 scans were acquired, alternating between the two CVs. In the case of 15 SPD and 30 SPD, precursors were isolated in a data-dependent manner using a 1.3 Th window, collected using automatic gain control (AGC) target value of 100% (1E5) and higher-energy collisional dissociation (HCD) fragmented at 28% normalized collision energy (NCE). Fragment spectra were acquired at a resolution of 15,000 at $m/z$ 200 using a maxIT of 25 ms. The precursor charge state filter was set to 2–6. The dynamic exclusion duration was set to 45 s and 90 s, adjusted to the gradient length of the 30 SPD and 15 SPD method. The cycle time was set to 2 s and 0.6 s for the 30 SPD and 15 SPD methods, respectively.

All settings are summarized in a tabular manner below.

## Database searching

### MQ searches

Peptide identification was done using MaxQuant (MQ) and its built-in search engine Andromeda (Cox and Mann, 2008). Each cohort was processed separately and searched against a canonical human reference database, including the peptide sequences of the PROCAL retention time standard peptides. Doubly injected cohort samples were assigned to the same experiment name. Raw files containing multiple internal CV values were split into separate files, each containing just a single CV. The latter were specified as different fractions of the respective cohort sample. Trypsin/P was set as the proteolytic enzyme allowing for up to two missed cleavages. Carbamidomethylated cysteine was considered a fixed modification and oxidation of methionine and N-terminal protein acetylation as variable modification. The match-between-runs (MBR) function was enabled with default settings. The Label-free quantification (LFQ) and the iBAQ option were disabled to reduce computation time and burden. Of note, the cohort searches could not be finished if LFQ normalization was activated. Both quantification metrics are calculated by the picked group tool (see below). The MQ searches were conducted employing a false discovery rate (FDR) of 100% for both peptide spectrum matches (PSMs) and on protein level.

The search results could either be directly subjected to protein grouping using the picked FDR method (The et al, 2022) or were Prosit rescored before (Gessulat et al, 2019). In either case, the 100% FDR evidence.txt files of each separate search had to be concatenated (no sorting necessary; only the columns "Sequence", "id", "Fraction", "Raw file", "Intensity", "Charge", "Experiment", "Mass", "Mass error [ppm]", "PEP", "Score", "Leading proteins", "Type", "Reverse", "Delta score", "Modified sequence", "MS/MS scan number", "Potential contaminant" are needed). The column "id" of the aggregated file was updated to 1 through the number of rows in the table, ensuring that each entry has a unique number.

### Prosit rescoring

Each MQ search was Prosit rescored separately (Gessulat et al, 2019). The resulting "prosit_target.psms" and "prosit_decoy.psms" files had to be concatenated before providing it as input to the picked group FDR tool (The et al, 2022). To this end, the files of each type (target or decoy) were concatenated, and the resulting files were sorted each by the posterior error probability ("posterior_error_prob") in ascending manner and by score ("score") in descending manner.

### Fragpipe searches

FragPipe version 21.1 was used for all searches with its built-in search engine MSFragger (Kong et al, 2017). Each cohort was processed separately. For the search without MSBooster (Yang et al, 2023) (FP_DDA), the "LFQ-MBR" workflow was used and the "Run MSBooster" option in the validation tab specifically unselected. For the search with MSBooster (FP_Booser) and for the wide-window mode search (FP_WWA) the "LFQ_MBR" workflow and the "WWA" workflow was selected, respectively. In the ".fp-maifest" files, raw files belonging to the same sample were given the same "Experiment" name and "Bioreplicate" number. Doubly injected cohort samples were not split in advance. The data type was set to "DDA" for FP_DDA and FP_Booster and to "DDA + " for FP_WWA. To receive results with 100% FDR as input for the picked group fdr tool, "--sequential --prot 1.0 --ion 1.0 --pep 1.0 --psm 1.0" was specified in the Filter field of the FDR Filter and Report section of the Validation tab and the "Print decoys" option was activated. All other options were kept as default.

### Picked group FDR tool

For all search strategies, the picked group FDR tool (The et al, 2022) was provided with the same canonical human reference FASTA file (downloaded 31.10.2023), including the peptide sequences of the PROCAL retention time standard peptides. Trypsin/P was specified as the protease for a full digest, up to two missed cleavages were allowed, the peptide length was set to 7–60 amino acids and the minimum number of peptides needed for LFQ calculation was set to 2.

For MQ-based searches, the picked group FDR tool was run on the combined 100% FDR "evidence.txt" file only (MQ) or including the combined, sorted "prosit_x.psms" files (MQ_Prosit).

For Fragger-based searches, the picked group FDR tool was run, providing the psm.tsv files of all samples and all combined_ion.tsv files of each separate cohort search as inputs.

For more detailed information, see https://github.com/kusterlab/picked_group_fdr.

## Data analysis

All data analysis was done using R in Rstudio with the use of the following packages clusterProfiler (v4.2.2), corrplot (v0.92), cowplot (v1.1.3), data.table (v1.15.0), doParallel (v1.0.17), dplyr (v1.1.2), ggbeeswarm (v0.7.2), ggplot2 (v3.4.3), ggrepel (v0.9.1), ggridges (v0.5.4), M3C (v1.8.0), parallel (v3.6.3), plotly (v4.10.3, rawDiag (v0.0.41), RColorBrewer (v1.1-3), stringr (v1.5.1), UpSetR (v1.4.0).

### Data filtering

While for MQ-based searches 1% FDR filtered files are provided as output by the picked group FDR tool, the resulting files for Fragger-based searches (named: "combined_protein.tsv") still had to be filtered to 1% FDR. Therefore, the posterior error probability (PEP) for each protein was calculated by subtracting the values of the "Protein Probability" column from 1 (1- "Protein Probability"). The table was sorted in ascending order based on the PEP and the q-value was calculated. The $q$-value for each row (protein) is represented by the mean PEP of all proteins with smaller or equal

PEP. All entries with *q*-values <= 0.01 were retained. For all searches, contaminants were removed before further data analysis.

### Overlaps protein grouping ambiguities

To calculate overlaps between proteins quantified by different search strategies and display them in an upset plot, differences in protein grouping between different searches had to be overcome. For protein groups with multiple Uniprot IDs in the MQ-based searches only the Uniprot ID that was the most frequently contained in all other searches was retained. In the Fragger-based outputs, protein groups with multiple Uniprot IDs did not exist

### Top abundant proteins

To filter the data to the top N most abundant proteins for the dimensional reduction visualization (UMAPs, Fig. 4B), the median iBAQ value for each protein across all samples of each cohort was calculated. The resulting values were given ranks per cohort. These ranks were used as cutoff (i.e., rank <= 10 for top-10 proteins).

### Imputation

Missing values have exclusively been imputed for the dimensional reduction visualization (UMAPs, Fig. 4B) which requires complete datasets. Imputation was only applied after filtering for proteins meeting the completeness cutoff (see below). Missing values were filled with values drawn from a random standard distribution in a protein wise manner. The simulated standard distribution had a median downshifted by 1.8 standard deviations from the median of the valid values of the respective protein and a standard deviation of 0.3 times that of the valid values.

### Defining a completeness cutoff

For each protein, a completeness ratio was calculated across all samples by dividing the number of quantified cases by the number of total cases. For each possible completeness cutoff (1–100%), the number of proteins meeting this criterion was counted. Plotting the number of proteins over the number of completeness results in a graph with three parts, a first steep section, a section with constant slope and another steep section. The first steep drop represents a bigger portion of proteins that are only identified in very few cases and the second steep drop represents a larger portion of proteins that are identified in almost all samples. To filter out the first part, we sought to define a cutoff that represents the start of the section with constant slope. Therefore, we approximated the first and second derivative of the relationship (*N* proteins over completeness, Fig. EV3C) described above. This approximation was achieved by taking the slope of linear regressions in sliding windows of 5 neighbors ($x - 2$ to $x + 2$, i.e., for 5% for the data points 3–7%). The completeness value at which the value of the second derivative first approaches 0 (change of slope = 0) represents the beginning of the linear section and was chosen as a completeness cutoff. In our data, this was the case at 13%, thus keeping proteins that were quantified in at least one cohort in 13% of cases.

### Defining a fold change cutoff

In order to define a meaningful threshold for calling differential protein expression, we randomly assigned samples of each cohort into two groups, keeping the number of samples of each cohort equal between the two groups. Calculating the (median) fold change for each protein between these groups showed a maximum absolute log2 fold change of 0.73 by chance alone. This maximum fold change observed by random chance alone gives insights into the variation in the dataset and allows to define a fold change cutoff for biologically relevant comparisons used for further analyses.

### Wilcoxon tests

For hypothesis testing based on protein LFQ intensities the pan-cancer cohort was split into two groups in an iterative way. For a given protein, the intensity values within a single cohort made up the first group and the intensity values of the same protein found in all other cohorts made up the second group (referred to as "mixed"). In each group the number of LFQ quantified values was at least 13% relative to the total number of cases, following the completeness cutoff rationale. *P* values were calculated by running the Wilcoxon rank test for each of the filtered proteins in each of the cohorts versus the respective the pan-cancer groups. Multiple testing correction was performed by applying the Benjamini–Hochberg *P* value correction. Proteins were considered significant, if their adjusted *P* value was below 0.01 as well as the log2 fold change greater than 0.73.

### Proteomics fingerprints

We defined classes in analogous fashion to the Human Protein Atlas project (HPA, (Uhlén et al, 2015)). Cohort-specific fingerprints contain three classes of proteins: Class I proteins are exclusively detected in one cohort only, Class II proteins are, on average, 1.66-fold (0.73 log2 fold; definition of cutoff see above) enriched over all other entities, and Class III proteins show 1.66-fold enhancement in one cohort compared to the average of all other cohorts (and not already covered in Class II). Proteins of any cohort falling into any of these three classes are considered as the cohort's specific fingerprint.

We compared these fingerprints to tissue-specific fingerprints defined by the HPA project (Uhlén et al, 2015) on RNA level. We chose healthy tissues representing the cancer tissue of origin as best as possible, being pancreas (for PDAC), brain (GBM), salivary gland/tongue (OSCC), stomach/ intestine (CRC), skin (MEL) and lymphoid tissue (DLBCL). Thereby dissecting the cohort-specific proteomic fingerprint into a tissue of origin-specific fraction (overlap to HPA data) and an entity-specific fraction.

### Enrichments analyses

GO term enrichments were done using the clusterProfiler R package (v4.2.2) (Yu et al, 2012) for all six cohorts for all three fingerprints on all ontology levels defining the whole *H. sapiens* database as background. The *P* value and *q*-value cutoff was set to 1. All results were combined, and a global adjustment of the *P* values was performed according to Benjamini–Hochberg.

For Hallmark enrichment analyses, we used the Molecular Signatures Database (msigDB) focusing on cancer hallmarks (Liberzon et al, 2015) as a reference database and performed Pearson's Chi-squared overrepresentation tests for each fingerprint and cohort. The results were combined, and *P* values were globally adjusted according to Benjamini–Hochberg. The five most significantly enriched hallmarks per cohort are displayed in Fig. EV5 and the rows are sorted by hierarchical clustering based on Jaccard similarity of the Hallmarks (genes contained in hallmark).

# Data availability

Raw data, the used FASTA files (including the FASTA file used for filtering out contaminants) and search engine output files have been deposited to the ProteomeXchange Consortium via the MassIVE partner repository with the dataset identifier MSV000095036. To enable easy and interactive exploration of the resource as well as additional analysis of the data by the community, we provide a custom-build Shiny App (https://panffpe-explorer.kusterlab.org/main_ffpepancancercompendium/) along with this manuscript. It enables the comparison of expression levels of all proteins or proteins in different categories (different classes of the cohort-specific fingerprint, oncogenes and tumor suppressors, differential variance and druggable proteins) across the six different entities or the quantitative differences of one cohort compared to the background of all other cohorts.

The source data of this paper are collected in the following database record: biostudies:S-SCDT-10_1038-S44318-024-00289-w.

# Peer review information

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

## Acknowledgements

The authors are grateful to L Lautenbacher, A Soleymaniniya and W Gabriel for providing access and help in order to test FragPipe on their Linux server. This work was partly funded by the German Federal Ministry of Education and Research (CLINSPECT-M grant no. FKZ161L0214A & 16LW0243K).

## Author contributions

**Johanna Tüshaus**: Conceptualization; Data curation; Software; Formal analysis; Validation; Investigation; Visualization; Methodology; Writing—original draft. **Stephan Eckert**: Conceptualization; Data curation; Software; Formal analysis; Validation; Investigation; Visualization; Methodology; Writing—original draft. **Marius Schliemann**: Data curation; Software; Formal analysis; Validation; Investigation; Visualization. **Yuxiang Zhou**: Resources; Data curation. **Pauline Pfeiffer**: Resources. **Christiane Halves**: Resources. **Federico Fusco**: Resources. **Johannes Weigel**: Resources. **Lisa Hönikl**: Resources. **Vicki Butenschön**: Resources. **Rumyana Todorova**: Resources. **Hilka Rauert-Wunderlich**: Resources. **Matthew The**: Software. **Andreas Rosenwald**: Resources. **Volker Heinemann**: Resources. **Julian Holch**: Resources. **Katja Steiger**: Resources. **Claire Delbridge**: Resources. **Bernhard Meyer**: Resources. **Wilko Weichert**: Conceptualization; Resources. **Carolin Mogler**: Resources. **Peer-Hendrik Kuhn**: Conceptualization; Resources; Supervision. **Bernhard Kuster**: Conceptualization; Formal analysis; Supervision; Funding acquisition; Validation; Investigation; Writing—original draft; Project administration.

Source data underlying figure panels in this paper may have individual authorship assigned. Where available, figure panel/source data authorship is listed in the following database record: biostudies:S-SCDT-10_1038-S44318-024-00289-w.

## Funding

## Disclosure and competing interests statement

BK is the founder and shareholder of OmicScouts and MSAID. He has no operational role in either company. The remaining authors declare no competing interests.

# Expanded View Figures

**Figure EV1.   Additional information provided by the TIC normalization approach.**

(**A**) Three MS1 TIC chromatograms of the same exemplary patient sample. Top: pre-analytical LC-FAIMS-MS run after the first sample preparation with low quality. Middle: pre-analytical LC-FAIMS-MS run after processing the sample a second time. Bottom: final analytical LC-FAIMS-MS/MS run of the reprocessed sample. (**B**) Scatter plot of the log10 sum of the MS1 TIC intensity of pre-analytical LC-FAIMS-MS runs as a function of the date of the first diagnosis as a proxy for the age of the processed FFPE sample. Each dot represents one sample from the melanoma cohort.

▶

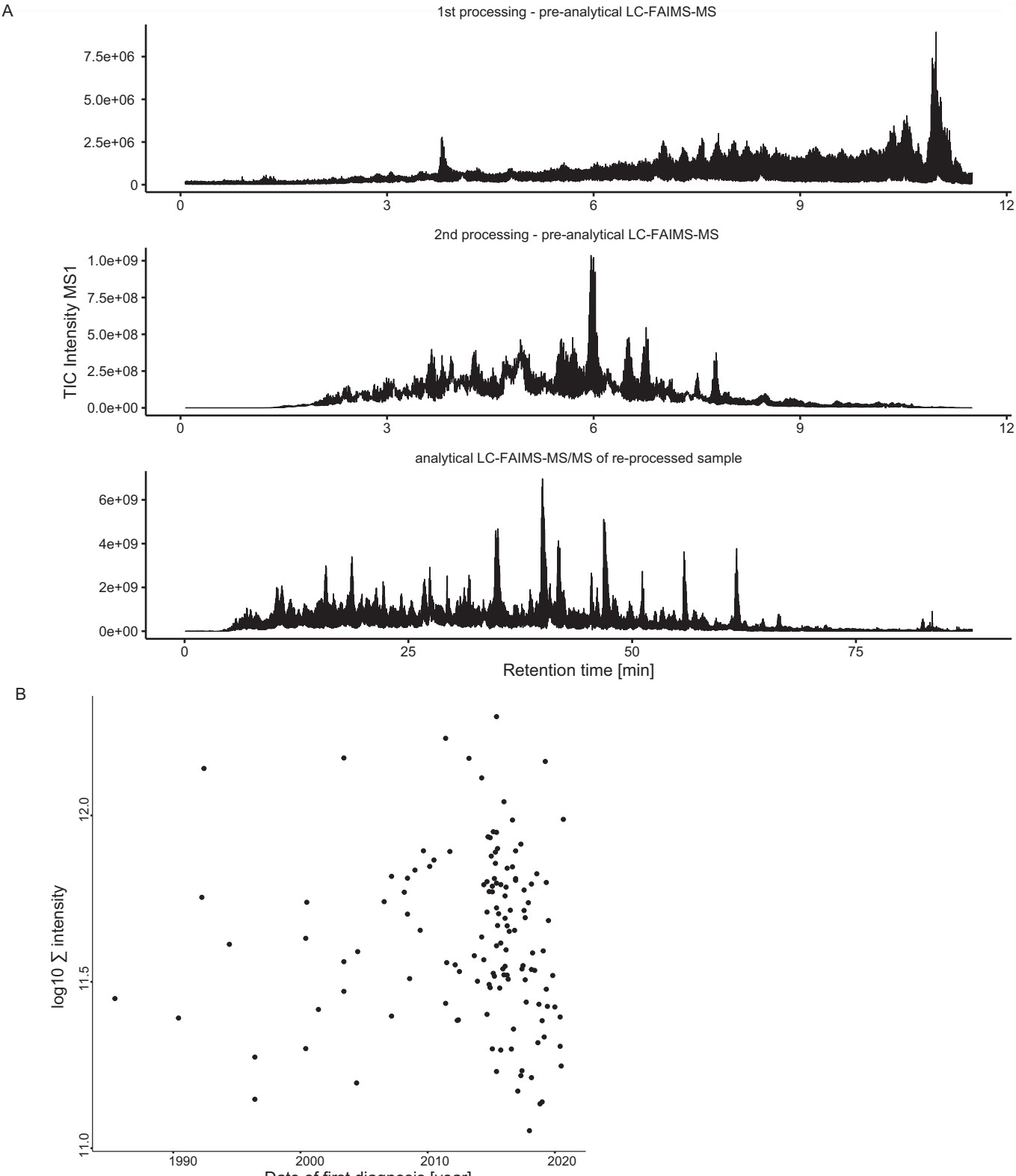

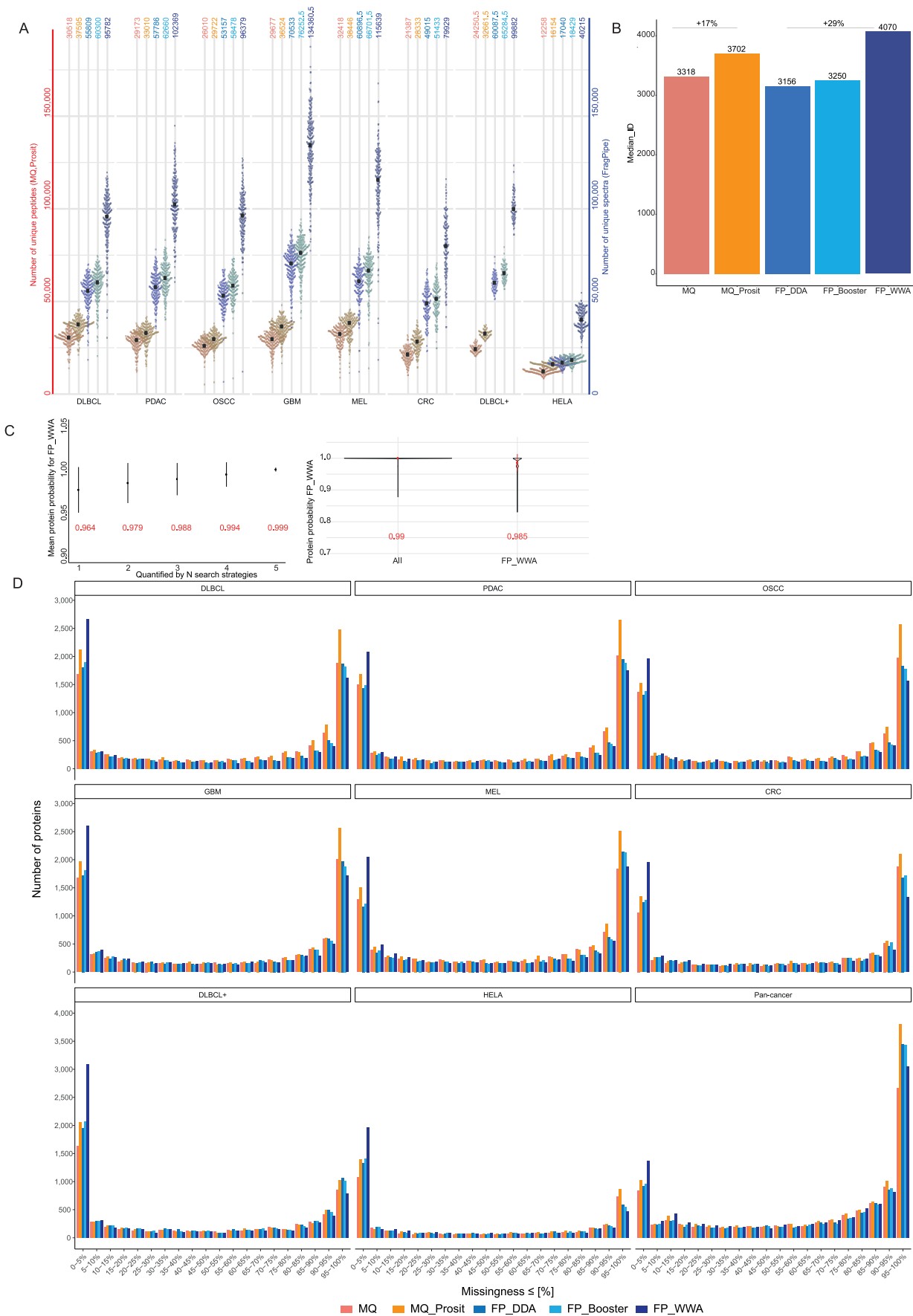

◀ **Figure EV2. Comparison of different search strategies for the analysis of the pan-cancer cohort.**

(A) Swarm plot indicating the number of unique peptides for the MaxQuant-based searches and the number of unique spectra for the FragPipe-based searches per FFPE tissue sample grouped by entity using different search strategies followed by picked protein group FDR in the following order MaxQuant (red), MaxQuant+Prosit (orange), FragPipe LFQ workflow without MSBooster (blue), FragPipe LFQ workflow with MSBooster (light blue) and FragPipe WWA (dark blue). (B) Bar plot showing the median number of quantified proteins per search strategy across all cohorts (HeLa excluded). The gains of post-processing are indicated in percent. (C) left: Dot whisker plot showing the mean protein identification probability for FragPipe WWA after picked group FDR for proteins as a function of the number of search strategies the protein was quantified in. 768, 565, 419, 774 and 8560 proteins were quantified by 1, 2, 3, 4 and 5 search engines, respectively. The whiskers represent the standard deviation. Right: Violine plots showing the distribution of the protein identification probability after picked group FDR for proteins that were quantified by all search strategies ($n = 8560$) vs. those quantified by FragPipe WWA but not quantified by all others ($n = 1631$). The red numbers and the red dot indicate the mean values, the whiskers the standard deviation. (D) Bar plots showing the number of missing proteins (in bins of 5%) for all five search strategies for all cohorts separately and all cohorts combined.

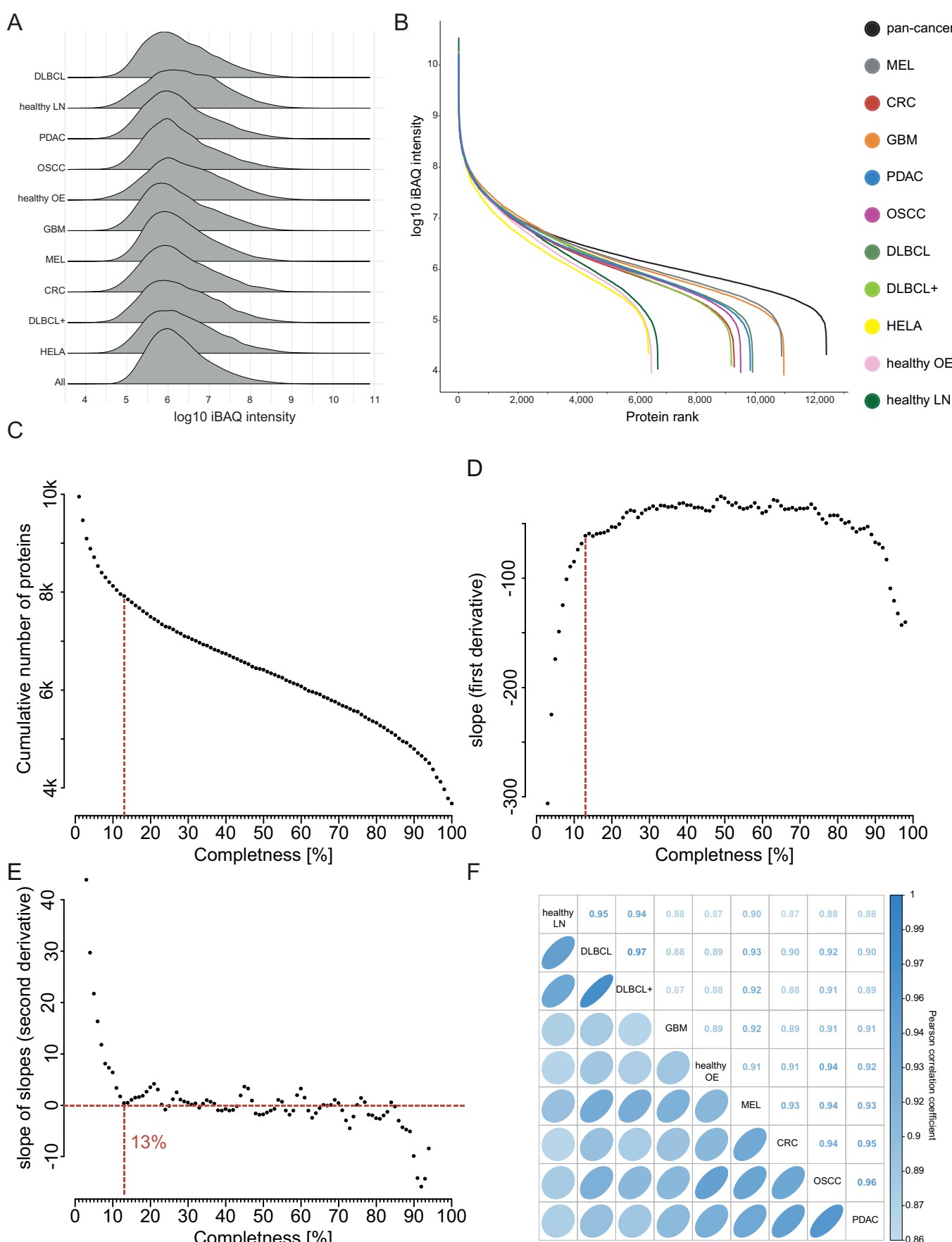

◄ **Figure EV3. Proteomic depth and definition of a completeness cutoff.**

(A) Ridge plots showing the distribution of the median log10 iBAQ values for all cohorts separately, HeLa QC samples and all cohorts combined (excluding HeLa samples). (B) Abundance rank plot of the median log10 iBAQ intensity of all iBAQ quantified proteins over the corresponding iBAQ Rank for each cohort separately and combined (excluding HeLa samples). (C) Dot plot showing the number of quantified proteins across all samples of all cohorts as a function of the completeness. The vertical, dashed line shows the chosen cutoff of 13%. (D) The approximated first derivative of the relationship displayed in (C). The vertical, dashed line shows the chosen cutoff of 13%. (E) The approximated second derivative of the relationship displayed in (C). The horizontal line highlights zero, no change in slope. The vertical, dashed line shows the chosen cutoff of 13%. (F) Correlation plot between cohorts and heathy tissue samples indicating the Pearson correlation coefficient. Cohorts are sorted by hierarchical clustering using Euclidean distance.

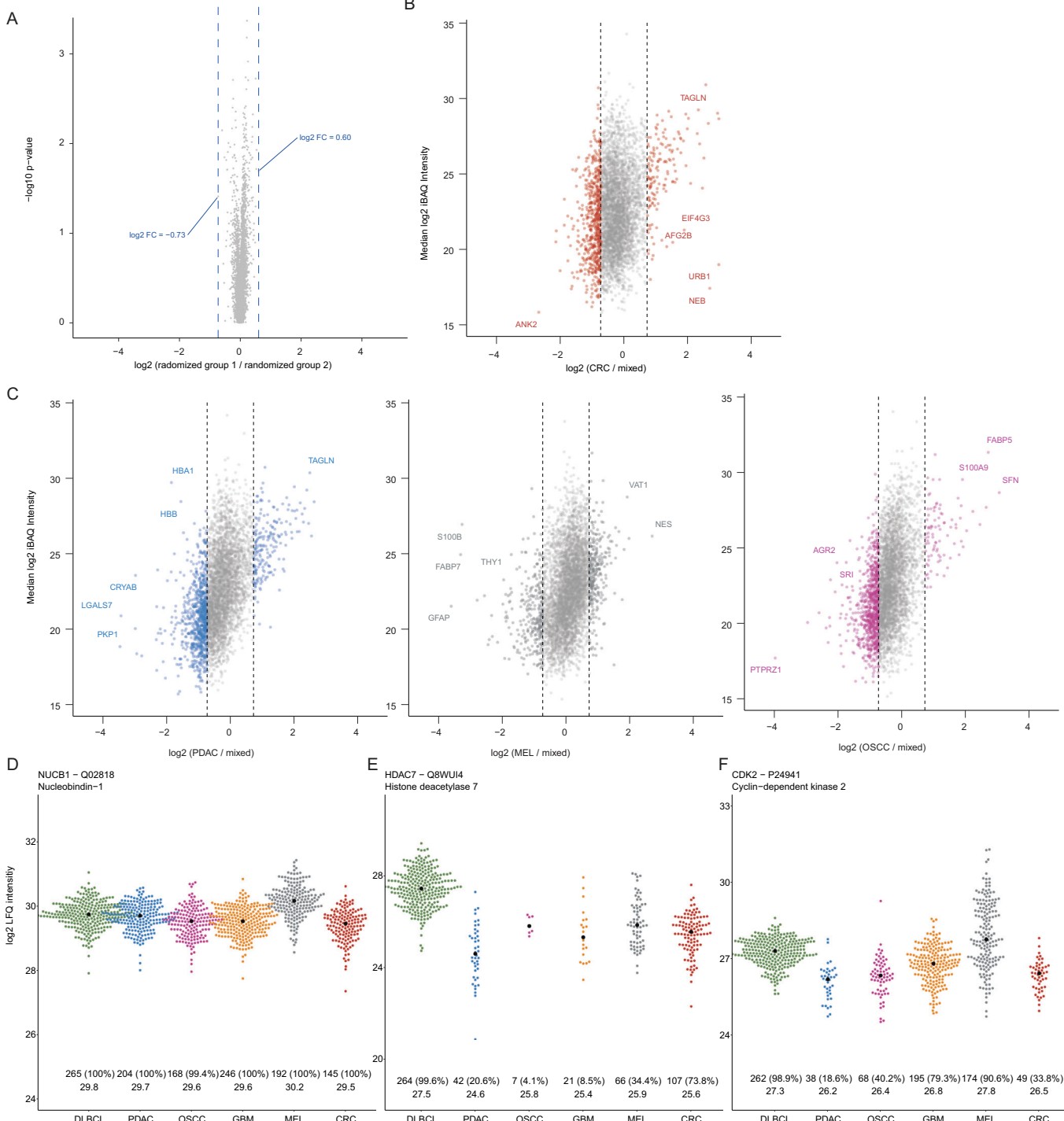

**Figure EV4.  Quantitative differences between cohorts.**

(A) Volcano plot showing the −log10 *P* value of the performed Wilcoxon's Rank test over the log2 fold change for all proteins between group1 and 2 (*n* = 609 patients each). Each patient sample was randomly assigned to one of the two groups, keeping the size of each cohort equal between the two groups. The maximum log2 fold change following this random assignment is indicated (blue). This maximum fold change observed by random chance alone gives insights into the variation in the dataset and allows to define a fold change cutoff for biological relevant comparisons used for further analyses. (B) Scatter plot comparing the expression of all proteins for CRC (*n* = 145) to the background of all other entities combined (*n* = 1075) using a Wilcoxon's Rank test. Each dot represents a protein. The log2 fold change of the median protein intensity for the respective entity vs the median protein intensity of all other entities is given on the *x* axes and the median log2 iBAQ intensity for the respective cohort is given on the y-axes. The dashed lines represent the fold change cutoff of ± 0.73 determined from (A). (C) Left: Same as (B) but for PDAC (PDAC: *n* = 204, combined background: *n* = 1,1016). Middle: same as (B) but for MEL. Right: same as (B) but for OSCC (*n* = 168, combined background: *n* = 1052). (D) Exemplary protein NUCB1 showing a rather stable degree of variability across all cohorts. The numbers at the bottom indicate the number of samples the protein was quantified in per cohort, the corresponding percentage and the median LFQ intensity. (E) Exemplary proteins HDAC7, druggable by small molecule inhibitors, enriched in DLBCL. (F) In contrast to NUCB1 in D) CDK2 exhibiting a higher degree of variability in MEL compared to other cohorts.

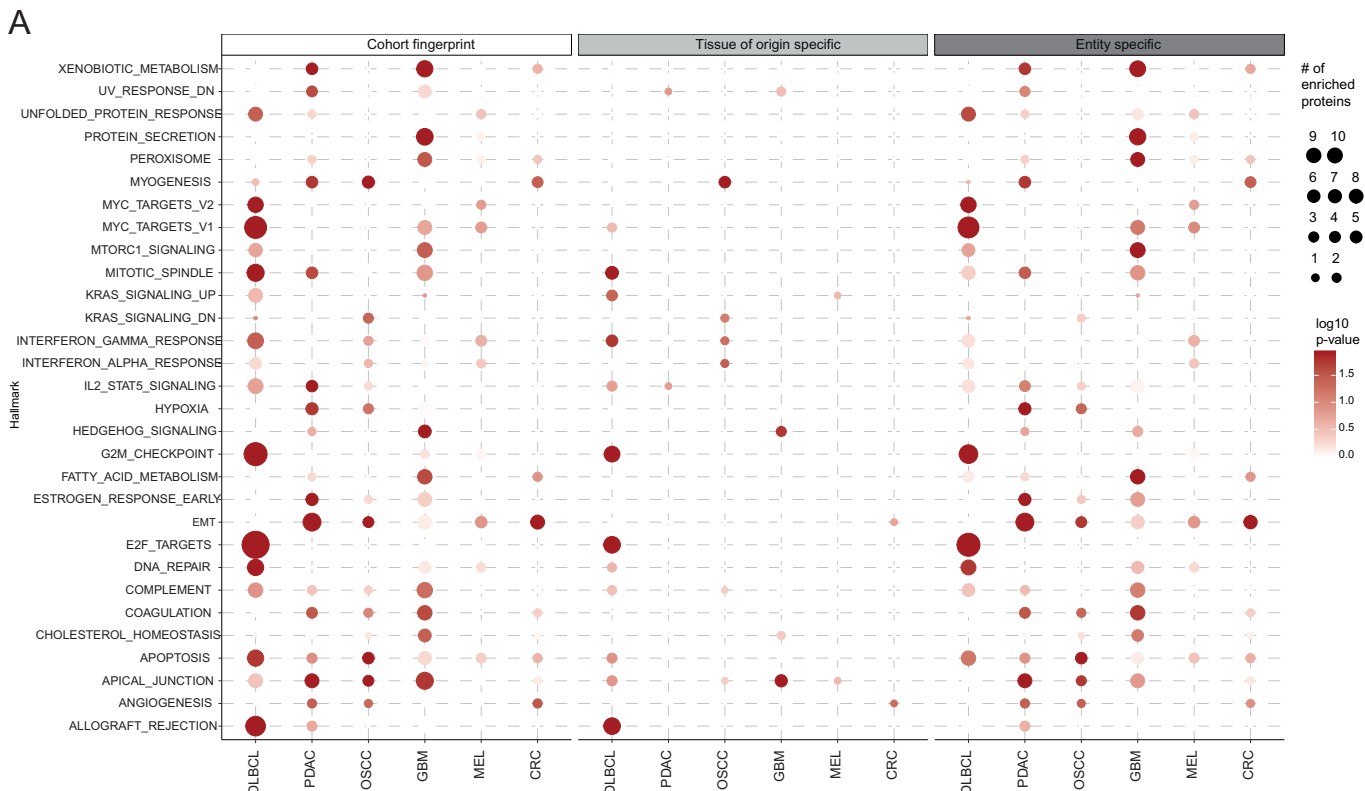

**Figure EV5. Hallmark of cancer enrichment analyses and comparison to healthy tissue.**

(A) Hallmark of cancer overrepresentation analysis based on a chi-squared contingency table test using the hallmark annotation database as background (MSigDB; Liberzon et al, 2015) for the cohort fingerprint, tissue of origin and cancer entity-specific proteins across all cohorts. The dot size represents the number of enriched proteins for the given Hallmark and the color scale indicates the statistical significance of the enrichment.

