## [Peer Review File · The EMBO Journal]

Towards routine proteome profiling of FFPE tissue: insights from a 1,220-case pan-cancer study

Johanna Tueshaus, Stephan Eckert, Marius Schliemann, Yuxiang Zhou, Pauline Pfeiffer, Christiane Halves, Federico Fusco, Johannes Weigel, Lisa Hönigl, Vicki Butenschön, Romyana Todorova, Hilka Rauert-Wunderlich, Matthew The, Andreas Rosenwald, Volker Heinemann, Julian Holch, Katja Steiger, Claire Delbridge, Bernhard Meyer, Wilko Weichert, Carolin Mogler, Peer-Hendrik Kuhn, and Bernhard Küster

Corresponding author(s): Bernhard Küster (kuster@tum.de)

Review Timeline:

Submission Date:	19th Jun 24
Editorial Decision:	31st Jul 24
Revision Received:	13th Sep 24
Editorial Decision:	29th Sep 24
Revision Received:	8th Oct 24
Accepted:	14th Oct 24

Editor: William Teale

Transaction Report:

Dear Dr. Küster,

Thank you again for the submission of your manuscript entitled "Towards routine proteome profiling of FFPE tissue: Insights from a 1,200 case pan-cancer study" and for your patience during the review process. We have now received the reports from two referees, which I copy below.

As you can see from their comments, both appreciated the clarity with which your data are presented and the methodological advance that was made in their generation. That said, some areas of the manuscript will require your attention before your manuscript can be published in The EMBO Journal.

Based on the overall interest expressed in the reports, I would like to invite you to address the comments of all referees in a revised version of the manuscript. I should add that it is The EMBO Journal policy to allow only a single major round of revision and that it is therefore important to resolve the main concerns at this stage. I believe the concerns of the referees are reasonable and addressable, but please contact me if you have any questions, need further input on the referee comments or if you anticipate any problems in addressing any of their points. I am always available to discuss the referee reports over Zoom, please let me know if you think this would be useful. Please, follow the instructions below when preparing your manuscript for resubmission.

I would also like to point out that as a matter of policy, competing manuscripts published during this period will not be taken into consideration in our assessment of the novelty presented by your study ("scooping" protection). We have extended this 'scooping protection policy' beyond the usual 3 month revision timeline to cover the period required for a full revision to address the essential experimental issues. Please contact me if you see a paper with related content published elsewhere to discuss the appropriate course of action.

Again, please contact me at any time during revision if you need any help or have further questions.

Thank you very much again for the opportunity to consider your work for publication. I look forward to your revision.

Best regards,

William

William Teale, Ph.D.
Editor
The EMBO Journal

When submitting your revised manuscript, please carefully review the instructions below and include the following items:

- 1) a .docx formatted version of the manuscript text (including legends for main figures, EV figures and tables). Please make sure that the changes are highlighted to be clearly visible.
- 2) individual production quality figure files as .eps, .tif, .jpg (one file per figure).
- 3) a .docx formatted letter INCLUDING the reviewers' reports and your detailed point-by-point response to their comments. As part of the EMBO Press transparent editorial process, the point-by-point response is part of the Review Process File (RPF), which will be published alongside your paper.
- 4) a complete author checklist, which you can download from our author guidelines ([https://wol-prod-cdn.literatumonline.com/pb-assets/embo-site/Author Checklist%20-%20EMBO%20J-1561436015657.xlsx](https://wol-prod-cdn.literatumonline.com/pb-assets/embo-site/Author%20Checklist%20-%20EMBO%20J-1561436015657.xlsx)). Please insert information in the checklist that is also reflected in the manuscript. The completed author checklist will also be part of the RPF.
- 5) Please note that all corresponding authors are required to supply an ORCID ID for their name upon submission of a revised manuscript.
- 6) We require a 'Data Availability' section after the Materials and Methods. Before submitting your revision, primary datasets produced in this study need to be deposited in an appropriate public database, and the accession numbers and database listed

under 'Data Availability'. Please remember to provide a reviewer password if the datasets are not yet public (see <https://www.embopress.org/page/journal/14602075/authorguide#datadeposition>). If no data deposition in external databases is needed for this paper, please then state in this section: This study includes no data deposited in external repositories. Note that the Data Availability Section is restricted to new primary data that are part of this study.

Note - All links should resolve to a page where the data can be accessed.

8) For data quantification: please specify the name of the statistical test used to generate error bars and P values, the number (n) of independent experiments (specify technical or biological replicates) underlying each data point and the test used to calculate p-values in each figure legend. The figure legends should contain a basic description of n, P and the test applied. Graphs must include a description of the bars and the error bars (s.d., s.e.m.).

9) We would also encourage you to include the source data for figure panels that show essential data. Numerical data can be provided as individual .xls or .csv files (including a tab describing the data). For 'blots' or microscopy, uncropped images should be submitted (using a zip archive or a single pdf per main figure if multiple images need to be supplied for one panel). Additional information on source data and instruction on how to label the files are available at .

10) We replaced Supplementary Information with Expanded View (EV) Figures and Tables that are collapsible/expandable online (see examples in <https://www.embopress.org/doi/10.15252/embj.201695874>). A maximum of 5 EV Figures can be typeset. EV Figures should be cited as 'Figure EV1, Figure EV2" etc. in the text and their respective legends should be included in the main text after the legends of regular figures.

12) Our journal encourages inclusion of *data citations in the reference list* to directly cite datasets that were re-used and obtained from public databases. Data citations in the article text are distinct from normal bibliographical citations and should directly link to the database records from which the data can be accessed. In the main text, data citations are formatted as follows: "Data ref: Smith et al, 2001" or "Data ref: NCBI Sequence Read Archive PRJNA342805, 2017". In the Reference list, data citations must be labeled with "[DATASET]". A data reference must provide the database name, accession number/identifiers and a resolvable link to the landing page from which the data can be accessed at the end of the reference. Further instructions are available at .

We realize that it is difficult to revise to a specific deadline. In the interest of protecting the conceptual advance provided by the work, we recommend a revision within 3 months (29th Oct 2024). Please discuss the revision progress ahead of this time with the editor if you require more time to complete the revisions. Use the link below to submit your revision:

Referee #1:

In general, this is a very well written manuscript reporting a proteome resource containing 1220 tumor proteomes, as well as lessons learned from the data acquisition and analysis. The key finding is the need of a new normalization method ensuring multi-batch proteome analysis. They also provided a nice web server for browsing the data. I have a few minor questions.

- 1, Abstract: "discovering that current software fails to process such large data sets": can you elaborate a bit what current software tools have you tried, and why they cannot analyze a data set of 1220 proteome with 2440 DDA runs? What are the software versions and what were the reasons? Out of RAM?
- 2, Introduction. Multiple studies have reported the proteome comparability of FFPE and non-FFPE tissues. Increasing studies analyze proteome of FFPE samples. The Introduction should not only narrow the focus on FFPE proteome studies. Studies using fresh frozen tissue samples, such as those pan-cancer studies from CPTAC, should also be cited and, ideally, compared with the data acquired from the current study.
3. More detailed clinical info of the patients from which the 1220 tumors were collected should be provided for query of the data.
- 4, How much ug FFPE tissue sample was processed for the proteome analysis? How about the % of tumor cells?
- 5, Can you please explain more details about the calibration curve of the Hela dilution? How could one borrow the TIC normalization method used in this study? How generic is it? Does it apply to DIA or TMT or SRM? Does it apply to other MS instruments? How frequently did you do this calibration?
- 6, Is there any difference in the protein/peptide missing rate when using the TIC MS1-only normalization strategy and the sample peptide normalization strategy? Is there any change in the number of proteins with at least two or more unique peptides?
- 7, In Figure 1A, the text and icons of the workflow do not align well.
- 8, In Figure 2B, there are points of different colors on the Hela dilution plot. What do they represent? What is the R2 of Hela dilution?

9, In Figure 2D, the "normalization run" and "analysis run" are shown in a boxplot with different colored borders, but the legend only shows blocks of color, which is a bit confusing.

Referee #2:

General summary

This manuscript describes the DDA proteomic analysis of 1,220 tumour samples (which are formalin-fixed paraffin-embedded, FFPE) from six cancer types processed over a period of three years. The authors demonstrate the robustness of the approach through this timeframe.

They also describe a new methodology for the normalisation of the samples to be analysed in the mass spectrometer, and then explain the need to separate the identification, quantification and FDR control stages in the bioinformatics analysis, since current software (they used MaxQuant and Fragpipe) cannot support the combined analyses of such big datasets, especially in this case since Ion mobility was used. Instead, they developed an alternative data analysis workflow using previously developed software in their group. Additionally, they perform rescoring of the identifications using the ProSight software. The authors describe the results of the proteomics profiling in some detail (although I am sure many more clinical conclusions can be extracted).

Overall, the study is well described and state-of-the-art, new methodology has been developed and all the results can be explored in a Shiny App.

Major and minor concerns

I have some suggestions though to make the manuscript even more informative:

- I don't think it is very clear from reading the manuscript (maybe only my impression) which are the proteins and their corresponding expression values that were retained in the final results: those ones identified by all methods (by at least two different peptides)?
- I think the authors should mention and discuss previous studies performed in FFPE tumour samples using transcriptomics approaches, highlighting also the limitations of those.
- Some extra information that could be added is how do the proteomics profiling results compare to the profiling of cell lines that are models for the tumour types included in the study? Does this comparison change significantly due to the FFPE embedded samples?
- Another aspect that it is not discussed is what would happen if such a study were performed in different labs, which is a direction of this type of studies in the future (due to e.g. the increase in number of samples). For instance, batch effects and comparability of iBAQ results across samples could be discussed.

Non-essential suggestions

- The authors could make easier the reuse of their bioinformatics workflow by third parties. There are different approaches, involved e.g. software containerisation and user documentation.
- I would have found interesting to search for both proteins and genes in the Shiny App. As far as I could see, one can only look for gene names at present.

Point-by-point response

The authors thank the reviewers for taking the time to evaluate our manuscript. We have prepared a detailed point-by-point response below and hope this clarifies all questions. In light of the valuable comments made, we have i) performed additional data analysis, ii) generated additional figures and iii) made changes to the text as indicated. The authors hope that the revised manuscript is now fit for publication.

Referee #1:

In general, this is a very well written manuscript reporting a proteome resource containing 1220 tumor proteomes, as well as lessons learned from the data acquisition and analysis. The key finding is the need of a new normalization method ensuring multi-batch proteome analysis. They also provided a nice web server for browsing the data. I have a few minor questions.

1. Abstract: "discovering that current software fails to process such large data sets": can you elaborate a bit what current software tools have you tried, and why they cannot analyze a data set of 1220 proteome with 2440 DDA runs? What are the software versions and what were the reasons? Out of RAM?

We modified the sentence of the abstract as follows to make the statement more precise.

"[...] Key findings include the need for a new normalization method ensuring equal and reproducible sample loading for LC-MS/MS analysis across cohorts, showing that tumors can, on average, be profiled to a depth of >4,000 proteins and discovering that current software fails to process such large ion mobility-based online fractionated data sets. [...]"

Each tumor sample was analyzed twice running an 88 min gradient with 5 different CVs each on an Evosep-FAIMS-Exploris 480 setup in data dependent acquisition (DDA) mode. In other words, we profiled 1220 tumor samples with 10 fractions each using online FAIMS-based fractionation. To profile the 1220 tumors, we performed 2440 LC-MS runs which result into 2440 raw files comprising 5 CVs each or after splitting the raw files according to their CVs - as required for MaxQuant analysis - a total of 12,200 split raw files.

We used two well-established search engines for the analysis of DDA data, namely MaxQuant and MSFragger. Detailed information regarding which software and version was used can be found in the methods section. MaxQuant was used alone as well as in combination with PROSIT rescoring. MSFragger was used alone, with MSBooster enabled or in wide window acquisition mode (WWA). Different versions of both software were tested including the latest released versions from 2024 but without success. When processing all 2,240 raw files including 5 CVs each (total of 1.76 TB) in a single large analysis, both search engines failed at the label free quantification step and the reason was likely that the system ran out of RAM. We tried the search on different workstations /clusters and even on the largest one (512 GB of RAM and 128 cores), the search never successfully finished. We are optimistic that both search engines could handle the analysis of 1220 single-shot LC-MS runs. However, difficult-to-analyze samples such

as FFPE tissue still require fractionation to enable deep proteome coverage which goes hand in hand with increasing computational burden during data analysis.

2. Introduction. Multiple studies have reported the proteome comparability of FFPE and non-FFPE tissues. Increasing studies analyze proteome of FFPE samples. The Introduction should not only narrow the focus on FFPE proteome studies. Studies using fresh frozen tissue samples, such as those pan-cancer studies from CPTAC, should also be cited and, ideally, compared with the data acquired from the current study.

We have expanded the introduction to highlight other pan-cancer studies, particularly the CPTAC program, that use fresh frozen tissue samples, as follows:

“[...] FFPE proteome profiling has been applied to the characterization of single cancer types such as colorectal adenomas (Coscia et al, 2020), lung cancer (Friedrich et al, 2021), ovarian tumors (Schweizer et al, 2023) and carcinoma of the esophagus (Li et al, 2023a). Most of these studies comprised relatively small cohorts of up to 100 cases. Larger entity-focused studies are beginning to emerge exemplified by the analysis of matched tumor and benign samples from 278 prostate cancer patients or the analysis of 1,780 thyroid nodules (malignant or not) to an average depth of 2,500 proteins each (Sun et al, 2022; Zhong et al, 2024). Larger-scale proteome profiling studies including thousands of patients of several entities have so far been limited to fresh frozen tumor samples, most prominently the CPTAC pan-cancer studies (Li et al, 2023b; Savage et al, 2024). Proteomic studies of fresh frozen tissue generally result in deeper proteome coverage and offer the possibility to analyze post-translational protein modifications, however, often suffer from limited specimen availability (Tushaus et al, 2023). To the best of our knowledge no large-scale pan-cancer FFPE study has been published yet. [...]”

To provide further context: the most recent pan-cancer study of the CPTAC consortium (Li et al., 2023b; Savage et al., 2024) covered ten entities of which four overlap with the six entities included in our pan-cancer FFPE project. GBM and PDAC were included in both studies. In addition, Head and neck squamous cell carcinoma (HNSCC) was included in the CPTAC study which encompasses OSCC included in our pan-cancer FFPE study. Moreover, CRC cancer included in our study, encompasses colon adenocarcinoma (COAD) which is included in the CPTAC study. We would like to point out that both proteomic profiling studies differ on many levels of the proteomic workflow making a direct comparison very difficult. These discrepancies start with the input material (fresh frozen vs FFPE), continue with the LC-MS setup used, the acquisition mode (label-free vs TMT) and end with the software tools utilized for data analysis – to name only the most relevant ones. Still, to answer the reviewer's comment, we downloaded the pre-processed CPTAC datasets of the four entities mentioned above from <https://pdc.cancer.gov/pdc/> and compared them to our proteomic data (Point-by-point response Figure 1). As shown below, the entity-specific overlap on protein level between the two studies is very large (60-68%) with only 3 to 10% of proteins identified exclusively in the panFFPE study. As expected, the CPTAC data exhibits higher proteome coverage because rather large quantities of fresh frozen input material were used and TMT-multiplexed protein digests were highly fractionated. Our workflow started with far less material, utilized FFPE tissue and

contains only an online fractionation step with FAIMS. We have added this comparison to the manuscript both in the form of text as well as new Appendix Figure S2.

Point-by-point response Figure 2: Venn diagrams illustrating the entity-specific overlap on protein level between our panFFPE study and the CPTAC pan-cancer (fresh-frozen, deep fractionated, TMT-multiplexed) study (Li et al., 2023b; Savage et al., 2024). Preprocessed data published by the CPTAC consortium were. The total number of patients is indicated as well as the number of overall identified proteins.

3. More detailed clinical info of the patients from which the 1220 tumors were collected should be provided for query of the data.

While we understand that this data would be valuable to share along with the proteomic data, we are not yet in a position to do so. This is because multiple follow-up studies and publications are underway that focus on querying the proteomic data using clinical data. These will be published separately and will include all available clinical data, which is why the current study has a more technical focus. Nevertheless, we added additional information to each tumor

sample to the Appendix of the paper stating inclusion criteria of the sub-cohorts, gender, Age at diagnosis and tumor cell count - as available (Appendix Table 1).

4. How much ug FFPE tissue sample was processed for the proteome analysis? How about the % of tumor cells?

We used approximately 50 µg protein of each FFPE tissue sample for the SP3-based protein cleanup and digestion approach. The percentage of tumor cells of each FFPE sample was added to the appendix of the paper (see comment above). The tumor cell content range between 15 and 98%, with a median of 80% across the pan-cancer cohort.

5. Can you please explain more details about the calibration curve of the HeLa dilution? How could one borrow the TIC normalization method used in this study? How generic is it? Does it apply to DIA or TMT or SRM? Does it apply to other MS instruments? How frequently did you do this calibration?

Generating a calibration curve is required for TIC normalization to be able to estimate and adjust sample quantities. Briefly, we used a tryptic HeLa digest of known concentration to perform a serial dilution covering 16 – 1,000 ng in 8 steps. These HeLa samples were loaded onto Evotips and analyzed using a 11.5 min LC gradient. The summed total ion current (TIC) intensity of all MS1 scans was calculated using the R script available at the Kuster's lab GitHub page (https://github.com/kusterlab/TIC_Normalization). Plotting the HeLa input amount over the summed MS1 TIC allowed us to derive a linear trend line equation. If the quality criterion of the HeLa dilution measurements was met ($R^2 \geq 0.97$), this calibration equation was used to calculate the concentration of peptide in the processed tumor samples.

The beauty of the TIC normalization method is that it is entirely generic. Because there is a strict linear relationship between the amount of peptide injected into the instrument and the MS signal it produces, the method is applicable to any type of MS instrument. For the same reason, this pre-analytical normalization step can be applied to any data acquisition mode including DDA, DIA, SRM, PRM and TMT data. Conceptually, TIC normalization is very similar to a Bradford or BCA protein/peptide assay that can be used to adjust protein amounts regardless of the sample processing protocol used downstream.

We repeated calibration curve generation for every batch (i.e. for every subcohort) for the full study. We advise to generate a new calibration curve for every sample batch because it will best reflect the current performance status of the mass spectrometer. The above information is already in the methods section of the manuscript but we made a small adjustment to the text as follows:

"[...] Key analytical findings include: i) demonstrating clear benefits by introducing a new, generic pre-analytical peptide quantification and normalization step to ensure equal and reproducible sample loading for LC-MS/MS analysis across cohorts; [...]"

6. Is there any difference in the protein/peptide missing rate when using the TIC MS1-only normalization strategy and the sample peptide normalization strategy? Is there any change in the number of proteins with at least two or more unique peptides?

To assess the influence of the two normalization strategies (Qubit and TIC) on the rate of missing data, we compared the seven samples which origin from the same FFPE GBM samples analyzed twice with input amounts adjusted according to the two methods (shown in Figure 2 C-D of the manuscript). Overall, TIC normalized samples are slightly more complete (15.5% of proteins detected in all samples) than Qubit normalized samples (11.5%; see Point-by-point response Figure 2, panel A below). Qubit-normalized samples, on average, led to slightly more protein identifications, but the variability of protein identification was smaller for TIC-normalized samples, which is why we chose TIC normalization for the pan-cancer cohort (Point-by-point response Figure 2, panel B below). This trend holds true for proteins identified with a minimum of 1 or 2 unique peptides. We added panel B of the point-by-point response Figure 2 below also to the main Figure 2 of the manuscript.

Point-by-point response Figure 3: Comparison of TIC or Qubit sample normalization using FFPE GBM samples (N=7). A) Bar graph indicating the completeness of the TIC vs Qubit normalized data (N=7). B) The determined standard deviation (SD) is consistently smaller for TIC than Qubit for the number of identified proteins with one or two unique peptides(right).

7. In Figure 1A, the text and icons of the workflow do not align well. We have re-aligned Figure 1A as follows:

8. In Figure 2B, there are points of different colors on the HeLa dilution plot. What do they represent? What is the R² of HeLa dilution?

The figure does not show actual data but is meant to illustrate the process of TIC normalization. Black points illustrate TIC measurements of controlled HeLa dilution samples to create a calibration curve. Colored lines illustrate measurements of patient samples and the matched colored points show the position of the patient samples on the calibration curve. We have adjusted the positioning of the legend to make this clearer.

We required the R² of the HeLa dilution to be better than 0.97 to ensure high quality of the TIC normalization procedure. If the R² of the HeLa dilution falls below 0.97, we advise preparing a fresh HeLa dilution curve and re-measure tumor samples if needed. We have added the following text to the Material and Methods section of the revised manuscript:

“ [...] For the TIC normalization approach, a small constant fraction of the total digest volume was loaded onto Evtips (see above) for each sample, along with an 8-point HeLa dilution series ranging from 16 ng to 1000 ng peptide per tip. The samples were analyzed via LC-FAIMS-MS using 2 CVs (-45 and -65 V) and an 11.5min (100 samples per day (SPD)) gradient (see Table 1 for details). As an approximation of the loaded peptide amount the total ion current (TIC) of all MS1 scans was summed up (equivalent to the area under the TIC chromatogram). Using this metric of the HeLa dilution series with known peptide amount a linear calibration curve was generated. R^2 of the HeLa dilution curve needed to be above 0.97 to ensure high quality. If this quality criterion were not fulfilled, measurements were repeated with a freshly prepared HeLa dilution series. Based on this regression the injected amount and in turn the concentration in the peptide digest of the FFPE samples was calculated. [...] “

- In Figure 2D, the "normalization run" and "analysis run" are shown in a boxplot with different colored borders, but the legend only shows blocks of color, which is a bit confusing.

We apologize for the confusion and removed the differently colored borders within the boxplot and adjusted the figure legend as follows:

Figure 2 D) Boxplots of the summed MS1 intensities of 11.5 min TIC normalization runs (intense color) and sample volume adjusted 88 min runs (pale color) for all cohorts. The numbers on top show the coefficient of variation (CoV) of the TIC sum for each cohort before and after normalization.

Referee #2:
General summary

This manuscript describes the DDA proteomic analysis of 1,220 tumour samples (which are formalin-fixed paraffin-embedded, FFPE) from six cancer types processed over a period of three years. The authors demonstrate the robustness of the approach through this timeframe.

They also describe a new methodology for the normalisation of the samples to be analysed in the mass spectrometer, and then explain the need to separate the identification, quantification and FDR control stages in the bioinformatics analysis, since current software (they used MaxQuant and Fragpipe) cannot support the combined analyses of such big datasets, especially in this case since Ion mobility was used. Instead, they developed an alternative data analysis workflow using previously developed software in their group. Additionally, they performed rescoring of the identifications using the Prosit software. The authors describe the results of the proteomics profiling in some detail (although I am sure many more clinical conclusions can be extracted).

Overall, the study is well described and state-of-the-art, new methodology has been developed and all the results can be explored in a Shiny App.

Major and minor concerns

1. I don't think it is very clear from reading the manuscript (maybe only my impression) which are the proteins and their corresponding expression values that were retained in the final results: those ones identified by all methods (by at least two different peptides)?

Apologies if this was not clear. To clarify, we tested different search strategies for our pan-cancer dataset. First, standard MaxQuant (MQ) (Cox & Mann, 2008), second a combination of MaxQuant with Prosit rescoring (MQ_Profit) (Gessulat *et al*, 2019), third standard FragPipe (FP_DDA), fourth FragPipe with MS Booster (FP_Booster) and fifth FragPipe with wide-window acquisition mode (FP_WWA) (Yang *et al*, 2023; Yu *et al*, 2021; Yu *et al*, 2023). Owing to the fact that FP_WWA outperformed all other search engines in terms of completeness and quantified protein groups per entity, we performed all subsequent data analysis using the FP_WWA results. To make this point clearer in the results part of the manuscript, we adapted the corresponding paragraph as follows:

"[...] More specifically, even the best-performing FP_WWA method only quantified 25% of all proteins in 77% of all samples (i.e. 23% missingness). All other methods showed substantially poorer performance using this metric. Due to the superiority of FP WWA in terms of completeness and identifications per cohort, all subsequent data analysis was performed using the FP WWA results. [...]"

2. I think the authors should mention and discuss previous studies performed in FFPE tumour samples using transcriptomics approaches, highlighting also the limitations of those.

While transcriptomic analysis of FFPE tumor samples is feasible in principle, it remains challenging because of degradation of mRNA during fixation (Jacobsen *et al.*, 2023; Kalmár *et al.*, 2015; Mirzazadeh *et al.*, 2023; Skaftason *et al.*, 2022). Truncated RNA, fragmented polyA tails, inefficient de-crosslinking and the procedure of the de-crosslinking step itself introduce substantial limitations. RNA degradation was suggested to be most severe during the first 6 months after fixation (Groelz *et al.*, 2018), on the contrary, chemically fixed proteins remain stable for years. Standard transcriptomic workflows have been adapted to overcome these challenges but still suffer from reduced transcriptome coverage and lower reducibility (Jacobsen *et al.*, 2023; Mirzazadeh *et al.*, 2023; Skaftason *et al.*, 2022). For example, a transcriptome study of 16 cases of DLBCL tumors was performed back to back with fresh frozen and FFPE material of the same tumor (Skaftason *et al.*, 2022). Fresh frozen samples revealed longer read lengths, higher reproducibility and higher number of total reads with a bias towards low abundant transcripts. Nevertheless, the paired fresh frozen and FFPE samples correlated extremely well with R^2 of 0.99 indicating the in general practicality and credibility the transcriptomic analysis of FFPE tissue (Skaftason *et al.*, 2022). Another study comparing transcriptomic data of paired FFPE and fresh frozen cardiac tissue reported a median RNA integrity number (RIN) of 8 for fresh frozen, while a median RIN of 2.5 was achieved for FFPE material (31%) (Jacobsen *et al.*, 2023). For FFPE tissue 48% of the reads mapped to protein coding genes, while 73% mapped for the fresh frozen material. Unmapped reads due to multimapping made up around 35% in the fresh frozen material and only about 18% in FFPE material. In contrast, unmapped reads due to shortness made up around 25% in FFPE and only about 3% in fresh frozen tissue (Jacobsen *et al.*, 2023).

We have added this information to the manuscript text as follows:

“[...] This may include the discovery or validation of molecular biomarkers by applying e. g. bulk or spatial omics analysis to large patient cohorts. Transcriptomic analysis of FFPE tissue is theoretically viable because high correlation has been observed in paired studies of FFPE and fresh frozen material, but actually remains challenging due to RNA degradation caused by the formalin fixation. RNA isolated from FFPE specimen exhibits lower quality (low median RNA integrity number) than fresh frozen material-derived RNA. This leads to shorter sequencing reads and, in turn, to a higher proportion of unmappable reads. (Jacobsen et al., 2023; Skaftason et al., 2022). Because most diseases manifest in altered proteome expression or activity and because most drugs act on proteins, it is conceptually attractive to analyze FFPE material at the proteome level (Coscia et al., 2020; Makhmut et al, 2023; Mund et al, 2022; Welker et al, 2015). [...]”

3. Some extra information that could be added is how do the proteomics profiling results compare to the profiling of cell lines that are models for the tumour types included in the study? Does this comparison change significantly due to the FFPE embedded samples?

To answer this question, we considered two published large-scale proteomic profiling studies of cancer cell lines. First, a study by the Gygi lab profiled 375 cell lines as part of the Cancer Cell Line Encyclopedia (CCLE) using TMT multiplexing (Nusinow *et al*, 2020). Second, the Australia-based ProCan team profiled 949 cancer cell lines from the Cancer Dependency Map project encompassing 40 cancer types using label-free quantification (Goncalves *et al*, 2022). The TMT study mapped proteomic differences between the cell lines relative to a control channel which enables comparison between the cell lines within their study but hampers a direct comparison to our study without reprocessing the data. Therefore, we decided to focus the comparison of our LFQ-based pan-cancer FFPE tumor tissue data to the LFQ study of ProCan/DepMap project.

173 cell lines represented the six entities contained in our study (based on TableS1 of Goncalves et al.). More specifically, 19, 13, 7, 36, 51 and 47 cell lines are associated to DLBCL, PDAC, OSCC, GBM, MEL and CRC, respectively. As a first approximation, to assess the overall correlation between cell lines and tumor samples, we compared the protein expression of the cell lines to the median protein LFQ value of all patients within a sub-cohort. For each ProCan/DepMap cell line and panFFPE entity combination, we calculated the Pearson's correlation coefficient and cosine similarity using the LFQ intensities. The correlation coefficients between tumor tissue and cell line ranged from 0.44-0.74, compared to 0.49-0.94 when comparing only cell lines to each other. Displaying the results in a heatmap, clustering the rows and columns based on Euclidean distance and annotating the rows based on cell line entity of origin, did not reveal clear grouping of cell line entities and patient entities (Point-by-point response Figure 4 A). For the heatmap showing the cosine similarity (row wise z-scored), the clustering is a little more pronounced for the DLBCL and MEL cases (Point-by-point response Figure 5 B) while none of the cell lines seem to correlate well to GBM.

Point-by-point response Figure 6 – Heatmaps depicting all pairwise comparisons between the 173 cell lines overlapping with the six FFPE sub-cohorts contained in our study based on entity of origin using Pearson's correlation coefficient (A) and z-scored cosine similarity (B). Columns and rows are clustered by Euclidean distance and rows are annotated based on entity of origin.

Patient tumors are molecularly homogeneous within a cancer entity and considering only the median LFQ value of each protein per sub cohort carries the risk to cover high correlating pairs of single patients to cell line. Therefore, in a next step we performed the same correlation analysis as described above, but for each patient to cell line pair separately (1220 patients x 173 cell lines) (Point-by-point response Figure 4). Protein intensity correlation between cell line and patient ranged between 0.2 up to 0.75. Again, no clear entity specific clustering is detectable and no high correlation pair (>0.9) was found. Overall many cell lines correlated well (around 0.5) to many patients independent of their entity status. Also, no cell line correlated extraordinary well to a sub-group of patients which would be an indicator as a good model system for that sub-group.

Point-by-point response Figure 7 – Heatmaps depicting all pairwise comparisons between the 173 cell lines overlapping with the 1220 patients included in our pan-cancer cohort using Pearson's correlation coefficient. Columns and rows are clustered by Euclidean distance and are annotated based on entity of origin.

Next, we plotted protein abundances of the best correlating cell line vs the median of a patient entity (Point-by-point response Figure 8) to illustrate that there is a clear positive correlation which, however, is not outstandingly strong. A combination of different factors likely causes the observed discrepancies. First, tumors are highly complex tissues and molecularly homogeneous even if classified as belonging to the same cancer type. These heterogeneity tumors can consequently not be represented well by a single cell line model. Second, the tissue samples comprise a mixture of cell types including potentially differently mutated cancer cells, immune cells, cell related to vasculature, connective tissue and many more. Many of these compartments are simply not present in a cultured cell line. Third, cell lines may harbor different driver mutations which can have profound influence on their proteomic profiles. Taken together, while cancer cell lines are often valid models of disease which come with a lot of benefits such as the ability to perform controlled experiments, they also suffer limitations such as they typically only represent one sub population of cancer driven by a specific oncogene and fail to represent the complex

architecture of tumor tissue. This limits the extent to which proteomic profiles of cell lines can be translated tumor biology. Due to the rather poor correlation scores, we decided not to include the cell line comparison results in the manuscript.

Point-by-point response Figure 8 – Scatter plots of the z-scored LFQ intensities of the best correlating model cell line to the z-scored median LFQ intensities of all patients of the correspond cancer entity contained in our study. Only pairwise complete observations are depicted, and each dot represents a protein. The number of proteins visualized in each plot and the number of exclusive proteins for cell line or FFPE cohort is given in the lower right corner of each plot. In the upper left corner of each plot, the R², Pearson’s correlation coefficient (rho) and the slope of the linear regression fitted to the data (blue dashed line) is given. The green dashed line represents the angle bisection (y=x).

4. Another aspect that it is not discussed is what would happen if such a study were performed in different labs, which is a direction of this type of studies in the future (due to e.g. the increase in number of samples). For instance, batch effects and comparability of iBAQ results across samples could be discussed.

This is clearly an important issue and experience from projects such as CPTAC shows that this is a substantial challenge. We have added the following text to the discussion section:

Further future directions in the overall workflow development may include improvements in sample throughput and sensitivity enabled by the latest generation of mass spectrometers (such as the Orbitrap Astral and timsTOF ultra (Ctorteccka et al, 2024; Heil et al, 2023) that may push the number of patient samples to >10,000 case per year per instrument and improve depth to >6,000 proteins per case. This would help to alleviate a substantial current challenge in clinical proteomics i. e. making sure that data can be collected at comparable depth and quality in different laboratories. Such multi-center studies are desirable in principle because the number and types of cancer cases that can be subjected to proteomic profiling may be increased. However, multi-center studies often suffer from the introduction of lab-specific batch effects and designing and enforcing standard operating procedures across sites remains difficult. An alternative approach would be to establish centralized laboratories at clinical centers with a particularly strong focus and expertise on certain cancer entities for an entire country or region. This would also require a way to coordinate the sample flow from decentralized biobanks to centralized expert-guided proteome centers.

Another area of future development is data consistency enabled by data independent acquisition (DIA). We consciously decided against a DIA approach at the beginning of the project for two main reasons. First, we expected that the overall proteome profiles of the pan-cancer cohorts would be quite different. Hence, it was unclear how one would design high quality spectral libraries that would adequately capture this anticipated (and now experimentally confirmed) heterogeneity and, at the same time, control for false positive assignments of proteins that, in fact, are only expressed in a subset of all samples and thus not missing for technical reasons. Again, the very fast and sensitive new generation mass spectrometers now enable narrow-window DIA measurements that diminish this particular issue. In addition, the scalability of DIA data analysis software was also unclear at the time and substantial improvements in DIA software performance are still needed in order to cope with the anticipated higher and higher data loads. Going forward, the authors think that proteome analysis software will have to move away from batch-wise processing of samples at the end of a project, but instead develop into a streaming mode of operation in which newly acquired data can be analyzed immediately, yet still aggregated as time progresses. As a result of project decisions taken several years ago, the current data set is not perfect in terms of proteomic depth (i.e. comprehensiveness) and data completeness across samples and cohorts, but it should be well controlled for false discoveries.

Non-essential suggestions

5. The authors could make easier the reuse of their bioinformatics workflow by third parties. There are different approaches, involved e.g. software containerisation and user documentation.

We agree with the reviewer that code accessibility is key for reproducing third parties work. In the context of this study, we have made the R script to perform the TIC normalization available on GitHub (https://github.com/kusterlab/TIC_Normalization). The PROSIT rescoring pipeline, called Oktoberfest, is also available via GitHub (<https://github.com/wilhelm-lab/oktoberfest>) and details are published by (Picciani *et al*, 2024). In addition, PROSIT rescoring is now integrated into MSFragger (<https://msfragger.nesvilab.org/>) which simplifies its use for many third party users. Instructions about using the picked group FDR tool, previously published by our lab (The *et al*, 2022) are also available at the Kuster laboratory GitHub page: https://github.com/kusterlab/picked_group_fdr. While we would love to be able to hand out our software in simple ways akin to what teams strongly focused on proteomic software development can do, we have to acknowledge our limitations in terms of human resources and software engineering skills available to our team. We, therefore, trust that interested parties will reach out to us and/or work with local experts to implement our tools.

For details on the wide window acquisition workflow of FragPipe, please, visit: (<https://fragpipe.nesvilab.org/>) (Yang *et al.*, 2023; Yu *et al.*, 2021; Yu *et al.*, 2023).

6. I would have found interesting to search for both proteins and genes in the Shiny App. As far as I could see, one can only look for gene names at present.

Based on this suggestion, we have updated the Shiny App to enables searching for gene names or UniProt IDs. Please, take a look at: https://panffpe-explorer.kusterlab.org/main_ffpepancancercompendium/

Coscia F, Doll S, Bech JM, Schweizer L, Mund A, Lengyel E, Lindebjerg J, Madsen GI, Moreira JM, Mann M (2020) A streamlined mass spectrometry-based proteomics workflow for large-scale FFPE tissue analysis. *J Pathol* 251: 100-112

Cox J, Mann M (2008) MaxQuant enables high peptide identification rates, individualized p.p.b.-range mass accuracies and proteome-wide protein quantification. *Nat Biotechnol* 26: 1367-1372

Ctortecka C, Clark NM, Boyle BW, Seth A, Mani DR, Udeshi ND, Carr SA (2024) Automated single-cell proteomics providing sufficient proteome depth to study complex biology beyond cell type classifications. *Nature Communications* 15: 5707

Friedrich C, Schallenberg S, Kirchner M, Ziehm M, Niquet S, Haji M, Beier C, Neudecker J, Klauschen F, Mertins P (2021) Comprehensive micro-scaled proteome and phosphoproteome characterization of archived retrospective cancer repositories. *Nat Commun* 12: 3576

Gessulat S, Schmidt T, Zolg DP, Samaras P, Schnatbaum K, Zerweck J, Knaute T, Rechenberger J, Delanghe B, Huhmer A *et al* (2019) Prosit: proteome-wide prediction of peptide tandem mass spectra by deep learning. *Nat Methods* 16: 509-518

Goncalves E, Poulos RC, Cai Z, Barthorpe S, Manda SS, Lucas N, Beck A, Bucio-Noble D, Dausmann M, Hall C *et al* (2022) Pan-cancer proteomic map of 949 human cell lines. *Cancer Cell* 40: 835-849 e838

Groelz D, Viertler C, Pabst D, Dettmann N, Zatloukal K (2018) Impact of storage conditions on the quality of nucleic acids in paraffin embedded tissues. *PLoS One* 13: e0203608

Heil LR, Damoc E, Arrey TN, Pashkova A, Denisov E, Petzoldt J, Peterson AC, Hsu C, Searle BC, Shulman N *et al* (2023) Evaluating the Performance of the Astral Mass Analyzer for Quantitative Proteomics Using Data Independent Acquisition. *bioRxiv*

Jacobsen SB, Tfelt-Hansen J, Smerup MH, Andersen JD, Morling N (2023) Comparison of whole transcriptome sequencing of fresh, frozen, and formalin-fixed, paraffin-embedded cardiac tissue. *PLoS One* 18: e0283159

Kalmár A, Wichmann B, Galamb O, Spisák S, Tóth K, Leiszter K, Nielsen BS, Barták BK, Tulassay Z, Molnár B (2015) Gene-expression analysis of a colorectal cancer-specific discriminatory transcript set on formalin-fixed, paraffin-embedded (FFPE) tissue samples. *Diagn Pathol* 10: 126

Li L, Jiang D, Zhang Q, Liu H, Xu F, Guo C, Qin Z, Wang H, Feng J, Liu Y *et al* (2023a) Integrative proteogenomic characterization of early esophageal cancer. *Nat Commun* 14: 1666

Li Y, Dou Y, Da Veiga Leprevost F, Geffen Y, Calinawan AP, Aguet F, Akiyama Y, Anand S, Birger C, Cao S *et al* (2023b) Proteogenomic data and resources for pan-cancer analysis. *Cancer Cell* 41: 1397-1406

Makhmut A, Qin D, Fritzsche S, Nimo J, König J, Coscia F (2023) A framework for ultra-low-input spatial tissue proteomics. *Cell Syst* 14: 1002-1014.e1005

Mirzazadeh R, Andrusivova Z, Larsson L, Newton PT, Galicia LA, Abalo XM, Avijgan M, Kvastad L, Denadai-Souza A, Stakenborg N *et al* (2023) Spatially resolved transcriptomic profiling of degraded and challenging fresh frozen samples. *Nature Communications* 14: 509

Mund A, Coscia F, Kriston A, Hollandi R, Kovács F, Brunner AD, Migh E, Schweizer L, Santos A, Bzorek M *et al* (2022) Deep Visual Proteomics defines single-cell identity and heterogeneity. *Nat Biotechnol* 40: 1231-1240

Nusinow DP, Szpyt J, Ghandi M, Rose CM, McDonald ER, 3rd, Kalocsay M, Jané-Valbuena J, Gelfand E, Schweppe DK, Jedrychowski M *et al* (2020) Quantitative Proteomics of the Cancer Cell Line Encyclopedia. *Cell* 180: 387-402.e316

Picciani M, Gabriel W, Giurcoiu VG, Shouman O, Hamood F, Lautenbacher L, Jensen CB, Müller J, Kalhor M, Soleymaniniya A *et al* (2024) Oktoberfest: Open-source spectral library generation and rescoring pipeline based on Prosit. *Proteomics* 24: e2300112

Savage SR, Yi X, Lei JT, Wen B, Zhao H, Liao Y, Jaehnig EJ, Somes LK, Shafer PW, Lee TD *et al* (2024) Pan-cancer proteogenomics expands the landscape of therapeutic targets. *Cell*

Schweizer L, Krishnan R, Shimizu A, Metousis A, Kenny H, Mendoza R, Nordmann TM, Rauch S, Kelliher L, Heide J *et al* (2023) Spatial proteo-transcriptomic profiling reveals the molecular landscape of borderline ovarian tumors and their invasive progression. *medRxiv*

Skaftason A, Qu Y, Abdulla M, Nordlund J, Berglund M, Ednersson SB, Andersson PO, Enblad G, Amini RM, Rosenquist R *et al* (2022) Transcriptome sequencing of archived lymphoma specimens is feasible and clinically relevant using exome capture technology. *Genes Chromosomes Cancer* 61: 27-36

Sun Y, Selvarajan S, Zang Z, Liu W, Zhu Y, Zhang H, Chen W, Chen H, Li L, Cai X *et al* (2022) Artificial intelligence defines protein-based classification of thyroid nodules. *Cell Discov* 8: 85

The M, Samaras P, Kuster B, Wilhelm M (2022) Reanalysis of ProteomicsDB Using an Accurate, Sensitive, and Scalable False Discovery Rate Estimation Approach for Protein Groups. *Mol Cell Proteomics* 21: 100437

Tushaus J, Sakhteman A, Lechner S, The M, Mucha E, Krisp C, Schlegel J, Delbridge C, Kuster B (2023) A region-resolved proteomic map of the human brain enabled by high-throughput proteomics. *EMBO J* 42: e114665

Welker F, Collins MJ, Thomas JA, Wadsley M, Brace S, Cappellini E, Turvey ST, Reguero M, Gelfo JN, Kramarz A *et al* (2015) Ancient proteins resolve the evolutionary history of Darwin's South American ungulates. *Nature* 522: 81-84

Yang KL, Yu F, Teo GC, Li K, Demichev V, Ralser M, Nesvizhskii AI (2023) MSBooster: improving peptide identification rates using deep learning-based features. *Nat Commun* 14: 4539

Yu F, Haynes SE, Nesvizhskii AI (2021) IonQuant Enables Accurate and Sensitive Label-Free Quantification With FDR-Controlled Match-Between-Runs. *Mol Cell Proteomics* 20: 100077

Yu F, Teo GC, Kong AT, Fröhlich K, Li GX, Demichev V, Nesvizhskii AI (2023) Analysis of DIA proteomics data using MSFragger-DIA and FragPipe computational platform. *Nature Communications* 14: 4154

Zhong Q, Sun R, Aref AT, Noor Z, Anees A, Zhu Y, Lucas N, Poulos RC, Lyu M, Zhu T *et al* (2024) Proteomic-based stratification of intermediate-risk prostate cancer patients. *Life Sci Alliance* 7

Dear Bernhard,

Thank you for submitting the revised version of your manuscript. It was sent to both reviewers; as you will see from the comments below, both are happy with the changes you made. Before I can formally accept your manuscript for publication, however, there are some remaining editorial points which need to be addressed. In this regard would you please:

- include a callout for Appendix Figure S3 in the main manuscript text,
 - complete 'sample definition' and 'in-laboratory replication' sections of the author checklist,
 - rename Appendix Table S1 as Dataset EV1 with the corresponding manuscript callout; upload legend as a separate tab in the Excel file,
- use the nomenclature 'Appendix Figure S1-S4' throughout the manuscript and Appendix PDF,
- indicate the statistical test used for data analysis in the legends of figures 6c; EV 4a-b; EV 5,
 - define box plots in terms of minima, maxima, centre, bounds of box and whiskers, and percentile in the legend of figure 2d,
 - define n in the legends of figures 2d; EV 2c; EV 4a-b,
 - define the measure of centre for error bars in the legend of figure 2c,
 - correct the section order as follows: title page with complete author information, abstract, keywords, introduction, results, discussion, methods, data availability section, acknowledgements, disclosure and competing interests statement, references, main figure legends, tables, expanded figure legends, and
 - check author emails for Johannes Weigl (johannes.weigl@tum.de), Marius Schliemann (marius.schliemann@tum.de) and Yuxiang Zhou (yuxiang.zhou@tum.de); these bounced when we tried them.

We include a synopsis of the paper (see <http://emboj.embopress.org/>). Please provide me with a two-sentence general summary statement and 3-5 bullet points that capture the key findings of the paper.

We also need a summary figure for the synopsis. The size should be 550 wide by [200-400] high (pixels). You can also use something from the figures if that is easier.

EMBO Press is an editorially independent publishing platform for the development of EMBO scientific publications.

Best wishes,

William

Dr William Teale
Editor
The EMBO Journal

Please remember: Digital image enhancement is acceptable practice, as long as it accurately represents the original data and conforms to community standards. If a figure has been subjected to significant electronic manipulation, this must be noted in the

figure legend or in the 'Materials and Methods' section. The editors reserve the right to request original versions of figures and the original images that were used to assemble the figure.

We realize that it is difficult to revise to a specific deadline. In the interest of protecting the conceptual advance provided by the work, we recommend a revision within 3 months (28th Dec 2024). Please discuss the revision progress ahead of this time with the editor if you require more time to complete the revisions. Use the link below to submit your revision:

Referee #1:

Thanks for the revisions. I have no more comments.

Referee #2:

I am happy with the rebuttal letter from the authors and I have no further comments. They have addressed very well all my previous comments. I think the manuscript has improved significantly in this revised version and it should be published.

All editorial and formatting issues were resolved by the authors.

Dear Bernhard,

I am pleased to inform you that your manuscript has been accepted for publication in the EMBO Journal.

Congratulations on publishing what I hope will be a really useful resource.

Best wishes,

William

William Teale, PhD
Editor
The EMBO Journal
w.teale@embojournal.org
